

# Dynamical spin-orbit-based spin transistor

**Fahriye N. Gürsoy[1], P. Reck[2], C. Gorini[2,3], K. Richter[2] and I. Adagideli[1,4,5*]**

**1** Faculty of Engineering and Natural Sciences, Sabancı University,
Orhanlı-Tuzla, 34956, Türkiye
**2** Institut für Theoretische Physik, Universität Regensburg, D-93040 Regensburg, Germany
**3** Université Paris Saclay, CEA, CNRS, SPEC, 91191, Gif-sur-Yvette, France
**4** TÜBİTAK Research Institute for Fundamental Sciences, 41470 Gebze, Türkiye
**5** MESA+ Institute for Nanotechnology, University of Twente,
7500 AE Enschede, the Netherlands

★ adagideli@sabanciuniv.edu

## Abstract

Spin-orbit interaction (SOI) has been a key tool to steer and manipulate spin-dependent transport properties in two-dimensional electron gases. Here we demonstrate how spin currents can be created and efficiently read out in nano- or mesoscale conductors with time-dependent and spatially inhomogeneous Rashba SOI. Invoking an underlying non-Abelian SU(2) gauge structure we show how time-periodic spin-orbit fields give rise to spin electric forces and enable the generation of pure spin currents of the order of several hundred nano-Amperes. In a complementary way, by combining gauge transformations with "hidden" Onsager relations, we exploit spatially inhomogeneous Rashba SOI to convert spin currents (back) into charge currents. In combining both concepts, we devise a spin transistor that integrates efficient spin current generation, by employing dynamical SOI, with its experimentally feasible detection via conversion into charge signals. We derive general expressions for the respective spin- and charge conductance, covering large parameter regimes of SOI strength and driving frequencies, far beyond usual adiabatic approaches such as the frozen scattering matrix approximation. We check our analytical expressions and approximations with full numerical spin-dependent transport simulations and demonstrate that the predictions hold true in a wide range from low to high driving frequencies.



# 1 Introduction

Following the theoretical proposal for a spin field-effect transistor by Datta and Das in 1990 [1], much research has focused on the realization of spin-based transistors that might eventually be more efficient and faster compared to the present transistors based on charge transport. The central issue for spin-based transistors is the realization of efficient generation, manipulation and detection schemes for spin currents and spin accumulations. While the conventional means of doing this involve ferromagnetic structures [2], these steps could also be achieved via all-electrical means, hence bypassing the need for magnetic components, by exploiting spin-orbit interaction (SOI). The proposals based on exploiting the phenomena of current-induced spin accumulation [3] and the spin Hall effect – the generation of spin currents transverse to

an applied electric field [4] – became the most commonly employed lines of study to achieve this goal.

The SOI can be induced by the electric fields of either charge impurities or the crystal lattice. In the latter case, one speaks of an "intrinsic" SOI. Prominent examples are Rashba- and Dresselhaus-type SOI in semiconductor quantum wells [5]. While the Dresselhaus SOI arises from a crystal's lack of inversion center, the Rashba term arises in low-dimensional systems, and it is induced by inversion symmetry breaking generated by the confining electric field of a quantum well. Because this electric field can be tuned by top or back gates, as established in two-dimensional electron gases (2DEGs) over 20 years ago [6], it is possible to modulate the Rashba SOI both spatially, and in principle, temporally. A spatial variation can generate a spin dependent force [7], and a temporally oscillating SOI may be capable of inducing charge and spin currents in the absence of any bias voltage [8–10]. Such physics is most conveniently handled by rewriting the SOI in terms of non-Abelian gauge fields [11–13]. Among other things, such an approach allows one to directly identify spin-electric and spin-magnetic fields [8, 14–17]. These fields accelerate electrons longitudinally or sideways à la Lorentz, respectively, but their sign depends on the electronic spin – opposite spin species are accelerated and bent in opposite directions. While these fields may exist in the presence of time-independent and homogeneous spin-orbit (and Zeeman) interaction, additional components appear when the SOI is time- and space-dependent [14, 15, 17]. The effects of driven or position dependent Rashba SOI have been further studied in various systems ranging from diffusive semiconductor [14, 18] and graphene [19–21] devices to spin pumping [22–27].

In this manuscript, we exploit the time- and space-tunability of Rashba SOI to explore how the non-Abelian (spin) gauge fields induce spin transport. We stress from the outset that the possibility of using time-dependent Rashba SOI for generating (i) AC spin currents in bulk diffusive systems [8, 28] and (ii) DC spin currents (hence spin pumping) in mesoscopic quantum wires [9] and quantum dots [10] has already been proposed. Moreover Onsager reciprocity relations have also been extended to spin dependent transport [16, 29, 30]. However, it was understood later on that in the presence of an additional non-Abelian gauge structure such as the one induced by the space- and time-dependent Rashba interaction, Onsager relations become more restrictive [17]. Here, we further explore the consequences of such "hidden" Onsager relations recently established for a spin and charge transport in 2DEGs. Among other things, our approach is applicable to diffusive and ballistic systems and is capable of handling effects of quantum coherence. Moreover, our method allows us to obtain general analytical formulae that go beyond the frozen scattering matrix approximation. [31, 32] –the conventional approach for spin pumping problems [33].

As a demonstration of our approach, we will devise a multi-terminal mesoscopic spin transistor, whose working principle builds upon the "hidden" Onsager relations [17]. While the transistor operates on pure spin currents, the output signal is a charge signal which can be detected by simple experimental procedures. The spin transistor setting, combining time-dependent and spatially-inhomogeneous SOI, is shown schematically in Fig. 1: The AC-modulated Rashba SOI on the right injects a pure spin current into the left region. The latter is then converted into a charge signal by a spin-magnetic field, induced by a static but non-homogeneous Rashba SOI, and read out as a voltage $V_{out}$ between contacts 1 and 3.

The paper is organized as follows. In Sec. 2, we introduce the model Hamiltonian and the non-Abelian gauge field approach, while in Sec. 3 we define the transport formalism, where we also discuss basics of the Floquet theory that will be necessary for our treatment. Our gauge field analytics are based on Ref. [17], allowing us to approximately express the desired spin-dependent quantities (time-dependent spin currents, AC spin conductances, etc.) in terms of conventional charge transport quantities (time-dependent charge currents, impedances, etc.), and to make general symmetry-based predictions concerning the expected output sig-

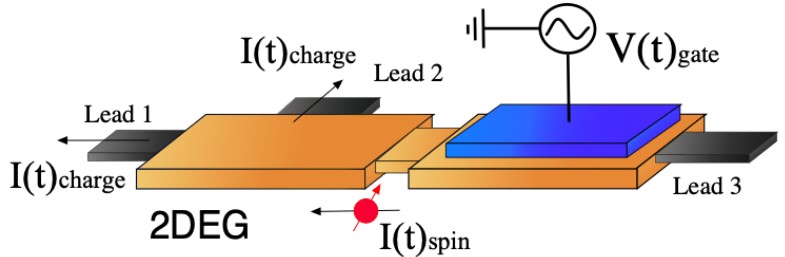

Figure 1: Sketch of the dynamical spin transistor. The on/off state of the spin transistor is controlled by pinching off Lead 2 using a (side) gate. A time-dependent top gate voltage generates a **pure** spin current in the right part of the device and injects it through the middle bridge.[1] This spin current is converted to a nonzero charge current on the left part in the multiterminal system via a spatially inhomogeneous Rashba interaction. Owing to the hidden Onsager relations, in a two-terminal system, the spin to charge conversion vanishes to leading order in the spin orbit coupling. This is the off-state of the transistor, which is achieved by pinching off the 2nd lead. The spin/charge conversion, hence the charge current out of the lead 1, is turned on by opening the second lead. This is the on-state of the transistor.

nals. We test such predictions against Floquet-based numerics which do not rely on any non-Abelian gauge manipulation, and which also allow us to go beyond analytically tractable low-frequency regimes. Thereby we demonstrate that our analytical expressions are applicable in a large range from low to high driving frequencies. In Sec. 4 we discuss spin current generation in a mesoscopic sample with homogeneous but time-dependent Rashba SOI, while in Sec. 5 we show how such spin signals can be non-locally converted into charge ones by the non-homogeneous but static Rashba SOI within a second mesoscopic cavity. This combined functionality is integrated and tested for a dynamical SOI-based spin transistor. Each section opens with low-frequency transport, before moving on to the high-frequency regime. Finally we present our conclusions in Sec. 6. To keep the manuscript as self-contained as possible, we gather a series of technical details in the appendices, where we also briefly discuss the generation of higher harmonics of spin and charge currents.

## 2 Model and its non-Abelian gauge structure

### 2.1 General case

Our starting point is the standard low-energy model for a spin-orbit coupled electron or hole gas in 2D [5]. The Hamiltonian reads

$$H = \frac{p^2}{2m} + \mathbf{b}(\mathbf{p}) \cdot \sigma + W(\mathbf{x}).$$ (1)

Here $W(\mathbf{x})$ is the electrostatic potential specifying the static environment, which might originate from gates, applied bias, impurities, etc., $m$ is the effective electron mass, and $\mathbf{b}(\mathbf{p})$ is a spin-orbit field, coupling spin to momentum. We assume that the strength of this SOI can be controlled externally and more generally, both in a position- and time-dependent way:

---

[1]The time-dependent potential $V(t)$ is discussed throughout our work in abstract terms as "a time-dependent gate potential", without specifying its source, e.g., electronic (GHz range) or optical (THz range).

$\mathbf{b}(\mathbf{p}) \to \mathbf{b}(\mathbf{p}; \mathbf{x}, t)$. For the rest of the paper, we specialize to electrons and light holes for which this field is linear in momentum. Then the Hamiltonian can be rewritten introducing non-Abelian gauge fields.[2] Following the notation of Ref. [17], the Hamiltonian reads

$$H = -\frac{D_\mu D_\mu}{2m} + V(\mathbf{x}). \tag{2}$$

Here $D_\mu = \partial_\mu - (i k_{so}/2)\sigma^a A_\mu^a$ is the $SU(2)$ covariant derivative along $\mu = x, y$, with $\sigma^a$ the Pauli matrices ($a = x, y, z$), $A_\mu^a(\mathbf{x}, t)$ a dimensionless $SU(2)$ vector potential and $V = W - k_{so}^2 (A_\mu^a A_\mu^a)/(8m)$. Unless specified otherwise, we assume that repeated indices are summed over. The SOI strength is controlled by the parameter $k_{so}$ which is typically much smaller than the Fermi momentum $k_F$. In terms of this spin-orbit parameter, the accuracy of Eq. (2) is $\mathcal{O}(k_{so}/k_F)^2$. We now make the central assumption that $L \ll l_{so}$, where $l_{so} = \pi/|k_{so}|$ is the spin-orbit length and $L$ the system size. This is usually fulfilled for experimental realizations using systems at nano- to mesoscales. We also assume that the time-dependence of $A_\mu^a(t)$ is slow, then $x_\mu k_{so} \partial_t A_\mu^a \ll \epsilon_F$, where $\epsilon_F$ is the Fermi energy. $SU(2)$ gauge transformations are unitary transformations of the form

$$U = \exp(i\Lambda_a(\mathbf{x}, t)\sigma_a/2). \tag{3}$$

It is then straightforward to show that this transformation maps $A_\mu^a(\mathbf{x}, t) \to (A')_\mu^a(\mathbf{x}, t)$ and $V(\mathbf{x}, t) \to V'(\mathbf{x}, t)$, where

$$(A')_\mu^a = A_\mu^a - \epsilon^{abc}\Lambda^b A_\mu^c + \frac{1}{k_{so}}\partial_\mu \Lambda^a, \tag{4}$$

$$V' = V - \sigma^a \frac{1}{2}\frac{\partial \Lambda^a}{\partial t}. \tag{5}$$

These transformations allow us, among other advantages, to gauge away the homogeneous and time-independent components of the spin-orbit field, up to quadratic order in the coupling constant [13, 37].

For a concrete example, we now specialize to a 2D electron gas with a Rashba spin-orbit interaction[3]

$$H = \frac{p^2}{2m} + \frac{1}{2}\left\{\alpha_R, \left(\sigma^x p_y - \sigma^y p_x\right)\right\} + W(\mathbf{x}), \tag{6}$$

where the Rashba coupling constant $\alpha_R$ can be a function of both position and time. However, for the sake of simplicity, we focus on regions of either time- or spatially dependent Rashba coupling, i.e. $\alpha_R(t), \alpha_R(\mathbf{x})$, and combine them later using rules for the combination of the respective scattering matrices. We identify and study two main effects, the generation of spin-electric and spin-magnetic fields to be discussed in the following two subsections.

## 2.2 Spin electric field from time-dependent SOI

A time-dependent Rashba SOI constant $\alpha(t) = k_{so} \sin(\Omega t)$, with $T = 2\pi/\Omega$ the AC modulation period from a top gate [6], will generate the spin dependent driving potential [8, 14, 16, 28], see Fig. 2.

In this case $A_\mu^a(t) = \epsilon_\mu^a \sin(\Omega t)$, and the $SU(2)$ gauge transformation (3) becomes

$$U = \exp\left(-i x_\mu k_{so} A_\mu^a \sigma^a/2\right). \tag{7}$$

---

[2]See e.g. Refs. [11, 12, 14–17]. Note that the representation in terms of non-Abelian gauge fields is actually exact, up to an irrelevant constant, if the spin-orbit field is constant in time and homogeneous in space.

[3]We note that the essential spin transport physics of a 2DEG with linear-in-momentum SOI is captured by Eq. (6), while details such as the precise spin polarization direction will depend on the presence and the strength of additional contributions, e.g. à la Dresselhaus.

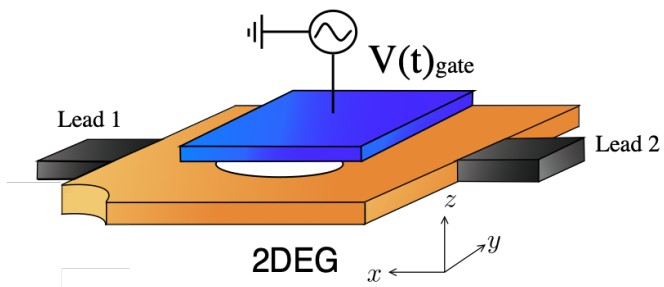

Figure 2: Sketch of the spin current source. A time-dependent top gate voltage is applied to a (chaotic) ballistic cavity connected to two leads, which controls the Rashba coupling and thus creates a spin electric force.

To order $(k_{so}L)^2$ one obtains a vanishing vector potential, while $\partial_t A_\mu^a(t)$ generates a non-zero $SU(2)$ scalar potential

$$(A')_\mu^a = 0\,, \tag{8}$$

$$(V')^a = x_\mu k_{so} \partial_t A_\mu^a \sigma^a / 2\,, \tag{9}$$

yielding the transformed Hamiltonian

$$H' = -\frac{1}{2m} \partial_\mu \partial_\mu + (x_\mu k_{so} \partial_t A_\mu^a) \frac{\sigma^a}{2} + V(\mathbf{x})\,. \tag{10}$$

A further global spin rotation [38] $\sigma^a \to \sigma^z$ then leads to the diagonal Hamiltonian

$$H_d = \frac{p^2}{2m} + V(\mathbf{x}) + \frac{V^s(t)}{2} \sigma^z\,, \tag{11}$$

$$V^s \equiv \epsilon_\mu^s x_\mu k_{so} \partial_t \sin(\Omega t)\,, \tag{12}$$

*i.e.* opposite-spin electrons feel a different electric field

$$\mathcal{E}_\mu^{\uparrow\downarrow} = -\partial_\mu [V(\mathbf{x}) \pm V^s(t)/2] \equiv E_\mu \pm \mathcal{E}_\mu^s\,. \tag{13}$$

The component $\mathcal{E}_\mu^s$ is a spin-electric field accelerating opposite spin species in opposite directions.

## 2.3 Spin-magnetic field from spatially inhomogeneous SOI

A spatially inhomogeneous Rashba SOI constant $\alpha_R(\mathbf{x}) = k_{so} \bar{\alpha}_R(\mathbf{x})$, with $\bar{\alpha}_R$ a dimensionless function, can either create spin currents or convert spin currents into charge currents, see Fig. 1. This interconversion is achieved by the spin-dependent Lorentz force due to a spin-magnetic field [17, 37]

$$\mathcal{B}^a = \partial_x A_y^a(\mathbf{x}) - \partial_y A_x^a(\mathbf{x})\,, \tag{14}$$

where $A_\mu^a(\mathbf{x}) = \bar{\alpha}_R(\mathbf{x}) \epsilon_\mu^a$. Below we specialize to the case

$$A_\mu^a = k_{so} \alpha_0 (\mathbf{x} \cdot \mathbf{f}) \epsilon_\mu^a\,, \tag{15}$$

with $\mathbf{f}$ a unit vector determining the fixed direction of the gradient of $\bar{\alpha}_R$. We now use the decomposition

$$A_\mu^a = -(\partial_\mu \chi^a + \epsilon_{\mu\nu} \partial_\nu \phi^a)\,, \tag{16}$$

for each spin component. Then we perform an $SU(2)$ gauge transformation

$$U = \exp(ik_{so}\,\chi^a\sigma^a/2)\,, \tag{17}$$

and consider effects to linear order in $k_{so}$. The vector potential becomes

$$(A')^a_\mu = \epsilon_{\mu\nu}\,\partial_\nu\,\phi^a\,, \tag{18}$$

$$(V')^a = V\,. \tag{19}$$

The transformed Hamiltonian becomes, up to linear order in the spin-orbit coupling,

$$H' = -\frac{1}{2m}[\partial_\mu + i\frac{k_{so}}{2}\epsilon_{\mu\nu}\partial_\nu\varphi(\mathbf{x})\sigma\cdot\mathbf{f}]^2 + V(\mathbf{x})\,, \tag{20}$$

where $\phi^a = \varphi(\mathbf{x})f^a$. We now perform a global spin rotation $\sigma\cdot\mathbf{f}\to\sigma^z$ to obtain

$$H_d = \frac{1}{2m}(\mathbf{p} + ik_{so}\mathbf{a}\,\sigma^z)^2 + V(\mathbf{x})\,, \tag{21}$$

where $\mathbf{a} = \alpha_0(\hat{\mathbf{z}}\times\mathbf{x})/2$. Hence the Hamiltonian structure implies that the two spin species decouple, each subject to a homogeneous magnetic field (as well as Lorentz force) of opposite sign. The strength of this magnetic field is given by $\sum_a\mathcal{B}^a f^a$, in agreement with Eq. (14).

# 3 Transport formalism: linear response and Floquet theory

## 3.1 Linear response currents

### 3.1.1 Basic expressions for charge and spin conductance

We consider a mesoscopic sample attached to leads labeled by $i, j$. The leads are in contact with different metallic reservoirs used to apply spin voltages. The linear response Landauer-Büttiker formula for the $a$-polarized spin current[4] flowing into/out of contact $i$ is generalized as follows [39, 40]

$$I_i^\alpha = \sum_\beta\left(\frac{2e^2}{h}N_i\delta_{\alpha\beta} - G_{ii}^{\alpha\beta}\right)V_i^\beta - \sum_\beta\sum_{j\neq i}G_{ij}^{\alpha\beta}V_j^\beta\,, \tag{22}$$

where $2N_i$ is the number of channels including spin in lead $i$. We use Latin letters (a=x,y,z) to denote the spin components of the current, while Greek letters commonly describe both electric (0) and spin components (a). Physically, the spin voltage $V_j^b = \mu_j^b/e$ represents an imbalance of spins polarized along a $b$-axis in reservoir $j$, with spin accumulation $\mu_j^b = \mu_j^{(\uparrow)} - \mu_j^{(\downarrow)}$. The conductance is a $(4\times 4)$-matrix in the combined spin-charge space:

$$G_{ij} = \begin{pmatrix} G_{ij}^{00} & G_{ij}^{0b} \\ G_{ij}^{a0} & G_{ij}^{ab} \end{pmatrix}\,, \tag{23}$$

where the superscript "0" indicates the charge component. Here $G \equiv G^{00}, G^{0b}, G^{b0}$ and $G^{ab}$ are, respectively, the charge, charge-spin, spin-charge and spin-spin conductances. They are

---

[4]Here we use the convention in which the physical dimension of the spin currents is the same as that of the charge currents, namely charge per unit time. The alternative, i.e. angular momentum per unit time, can be obtained by multiplying with the factor $\hbar/2e$.

defined as

$$G_{ij}^{00} = \frac{e^2}{h} \sum_{m,n} \text{Tr}[t_{mn}^\dagger t_{mn}], \tag{24}$$

$$G_{ij}^{0b} = \frac{e^2}{h} \sum_{m,n} \text{Tr}[t_{mn}^\dagger t_{mn} \sigma^b], \tag{25}$$

$$G_{ij}^{b0} = \frac{e^2}{h} \sum_{m,n} \text{Tr}[t_{mn}^\dagger \sigma^b t_{mn}], \tag{26}$$

$$G_{ij}^{ab} = \frac{e^2}{h} \sum_{m,n} \text{Tr}[t_{mn}^\dagger \sigma^a t_{mn} \sigma^b]. \tag{27}$$

Here, $t_{mn}$ is the (2×2)-matrix of spin dependent transmission amplitudes connecting channel $n$ in lead $j$ to channel $m$ in lead $i$, and $\sigma^a$ are the Pauli spin matrices.

Even in the absence of any charge bias the system can feature a charge response given by

$$I_i^0 = -\sum_b G_{ii}^{0b} V_i^b - \sum_b \sum_{j \neq i} G_{ij}^{0b} V_j^b, \tag{28}$$

in addition to the spin response

$$I_i^a = \sum_b \left( 2\frac{e^2}{h} N_i \delta_{ab} - G_{ii}^{ab} \right) V_i^b - \sum_b \sum_{j \neq i} G_{ij}^{ab} V_j^b. \tag{29}$$

### 3.1.2 Link between spin and charge conductance via the gauge transformation

We now use the gauge transformations discussed in Sec. 2, to obtain an expression for the spin conductances. As discussed above, we consider the case of time-dependent SOI and position dependent SOI separately. After the gauge transformation and working up to linear order $k_{so}$, the full Hamiltonian is block-diagonalized into spin blocks. The time-dependent part of the SOI transforms into spin-dependent electric fields, which we treat as spin-dependent voltages $V^a = V^\uparrow - V^\downarrow$. The position dependent part of the SOI enters as a spin-dependent magnetic field as shown in Eq. (21). Owing to the block diagonal structure, the spin conductance in the transformed basis is simply

$$G_{ij}^a = G_{ij}^\uparrow - G_{ij}^\downarrow. \tag{30}$$

We now transform back to the initial gauge to obtain the spin-conductances in the original spin basis:

$$G_{ij}^a \approx \left[ G_{ij}(\mathcal{B}_z^a) - G_{ij}(-\mathcal{B}_z^a) \right] f^a. \tag{31}$$

Here $G_{ij}(\mathcal{B}_z^a)$ is the charge conductance of a spinless electron moving in the same scalar potential $V$ (including confinement and disorder) as the original electron, but in the presence of an additional induced magnetic field $\mathcal{B}_z^a$ given in Eq. (14). Thus the gauge transformation allows us to express the mesoscopic spin conductances in terms of charge conductances.

Using Onsager relations and current conservation, one can show that $G_{ij}^a$ in Eq. (31) vanish in a system with 2 connected leads in the presence of time reversal symmetry. Under the assumption $L \ll l_{so}$ where $l_{so} = \pi/|k_{so}|$ – i.e. a moderate Rashba constant $k_{so}$ – the spin precession conductance $G^{ab}$ is equivalent to the charge conductance, $G^{ab} \propto \delta^{ab} G^{00}$. Hence all spin conductances in Eqs. (28) and (29) can be expressed in terms of the charge conductance to linear order in $k_{so}$.

In our full numerical calculations below we will compute the driven charge and spin currents without resorting to the approximate $SU(2)$ manipulations described here and in Sec. 2.

Using these numerics as reference calculations, we will be able to show that the description in terms of spin-electric and spin-magnetic fields acting on effectively decoupled ↑↓-electrons is very accurate up to $L \sim l_{so}$ although Eq.(31) is valid formally for small $k_{so}$.

## 3.2 Floquet scattering theory

### 3.2.1 General framework

When interacting with a dynamical scatterer with oscillation frequency $\Omega$, an electron impinging at energy $E$ can absorb or emit an integer amount of energy quanta $n\hbar\Omega$, leaving the scattering region with energy $E_n = E + n\hbar\Omega$ ($n = 0, \pm 1, \pm 2, \dots$). The scattering matrix then depends on the initial and final energies $E$ and $E_n$, and is referred to as the Floquet scattering matrix, $S_{F,im,jm'}(E_n, E)$ [41]. This matrix gives the amplitude for a process where an electron from channel $m'$ in lead $j$ at energy $E$ is scattered into channel $m$ within lead $i$ at energy $E_n$.

We now calculate the Floquet scattering matrix. To this end, one has to numerically solve the time-dependent Schrödinger equation. As reviewed in Appendix B, Floquet theory allows one to express the full-time evolution in terms of an effective static matrix Hamiltonian, expressed in terms of the Floquet states $|n\rangle$,

$$H_F = \sum_{n=-\infty}^{\infty} \left( (H_0 - n\hbar\Omega)|n\rangle\langle n| + \frac{iH_1}{2}(|n\rangle\langle n+1| - |n\rangle\langle n-1|) \right). \tag{32}$$

Here, $H_0 = -\frac{\hbar^2}{2m}(\partial_x^2 + \partial_y^2)$ and $H_1 = ik_{so}(\sigma_y\partial_x - \sigma_x\partial_y)$ are the kinetic and the Rashba contributions – recall from Sec. 2 that the time-modulated Rashba SOI is homogeneous, *i.e.* $\alpha_R = \alpha_R(t)$. The Floquet states $|n\rangle$ are defined as the basis vectors of the periodic eigenfunctions $|\phi_\eta(\vec{r}, t)\rangle$ of the Floquet Hamiltonian

$$|\phi_\eta(\vec{r}, t)\rangle = \sum_n e^{-in\Omega t}|n\rangle. \tag{33}$$

For numerical calculations one has to truncate the (infinite) sum over Floquet bands $n$. In Appendix F we show that including up to 21 bands ($|n| \leq 10$) provides enough accuracy for our purposes. After obtaining the elements of the Floquet scattering matrix, we calculate the AC currents in the absence of a bias voltage. The charge/spin currents are expressed as

$$I_i^\alpha(t) = \sum_{l=-\infty}^{\infty} e^{-il\Omega t} I_{i,l}^\alpha. \tag{34}$$

The Fourier components are then determined in terms of the Floquet scattering matrices $S_F(E_n, E)$ as [32]

$$I_{i,l}^\alpha = \frac{e}{h}\frac{\hbar}{2} \int dE \sum_{n=-\infty}^{\infty} \sum_{j=1}^{N_r} \sum_{m\in i, m'\in j} Tr[S_{F,im,jm'}^\dagger(E_n, E)\sigma^\alpha S_{F,im,jm'}(E_{l+n}, E)], \tag{35}$$

where $N_r$ denotes the number of leads, and $\sigma^0 = 1$. We refer the readers to Appendix D for further details.

To obtain the dynamical scattering amplitude and determine the spin and charge currents, we use the tight binding form of the Floquet Hamiltonian (32) (see Appendix A) and calculate the scattering matrix using the software package Kwant [42]. We consider only $l = \pm 1$ for the AC response.

We will consider a wide range of frequencies, numerically probing both the low-frequency ($\Omega \ll 1/\tau$) and high-frequency regimes ($\Omega \gtrsim 1/\tau$), where $\hbar/\tau$ is the typical internal energy

scale of the scattering matrix. For few channel ballistic transport, $\tau$ is given by the time of flight of an electron between two leads, which is calculated using the Wigner-Smith time-delay matrix [43, 44] as shown in Appendix E. If the frequency is much smaller than the inverse of the time of flight and the Fermi energy $\epsilon_F$, $\Omega\tau \ll 1$ and $\hbar\Omega \ll \epsilon_F$, the scattering process is in the adiabatic limit.

### 3.2.2 Floquet theory in the adiabatic limit

In the adiabatic limit, the calculation of the Floquet scattering matrix can be simplified as was suggested in Refs. [31, 32, 41]. Considering a set of time-dependent parameters $p_\alpha(t)$, one calculates a stationary scattering matrix with these parameters at a given time $t$. This scattering matrix is called the frozen scattering matrix:

$$S(E, t) = S(E, \{p_\alpha(t)\}). \tag{36}$$

For cases of interest here, these parameters $p_\alpha(t)$ are periodic in time with period $\mathcal{T} = 2\pi/\Omega$. In the adiabatic limit, $\Omega\tau \ll 1$ and $\hbar\Omega \ll E_F$, the full Floquet scattering matrix can then be approximated as follows:

$$S_F(E_n, E) \simeq \frac{1}{\mathcal{T}} \int_0^{\mathcal{T}} S(E, t) e^{in\Omega t} dt. \tag{37}$$

This is called the frozen scattering matrix approximation of the Floquet scattering matrix.

We compute the adiabatic approximation (37) of the Floquet scattering matrix and check the limits of its validity in Appendix C.

## 4 Spin current generation

We first study spin current generation from a spin-dependent voltage using Eq. (29) in both the adiabatic and high-frequency regimes. We consider a 2D chaotic cavity with only time-dependent Rashba coupling, induced by an AC top gate voltage, see Fig. 2. The Rashba SOI strength has a time dependence given by $\alpha_R(t) = k_{so} \sin(\Omega t)$, with $\Omega$ the driving frequency. As discussed above and previously shown in references [8, 14, 16, 28], this time-dependent coupling induces a spin voltage on the leads, and the spin voltage difference between two leads results in a spin current polarized in the $y$-direction. Throughout this section, we will be working in the linear response regime and ignore effects nonlinear in the spin-orbit coupling. The general spin current formulae are derived in Secs. 4.1 and 4.2. The numerical results follow in Sec. 4.3, also including a comparison with our analytical results in Sec. 4.4.

### 4.1 AC spin current in the low-frequency regime

In the low-frequency regime, one can neglect the frequency dependence of the charge conductance. The spin current in Eq. (29) can thus be computed using the DC conductance. To leading order in the spin-orbit coupling, all spin effects are included in the spin voltage. Hence $G^{ab} = \delta^{ab}G$.[5] In the absence of a bias voltage ($V^0 = 0$), the spin current at lead 1 is the current generated by the spin voltage

$$I_1^a = \sum_j \left( 2\frac{e^2}{h}N_1\delta_{1j} - G_{1j} \right) V_j^a. \tag{38}$$

---

[5]We note that the case of a large Rashba coupling with a small time-dependent part can also be treated with our gauge transformation method. However if $l_{so} \sim L$ the results of this section need be modified to account for spin precession.

The spin voltage from the time-dependent Rashba SOI reads

$$V^{a,\uparrow(\downarrow)} = \pm\partial_t\alpha_R(t)\epsilon^a_\mu l_\mu\,. \tag{39}$$

Here $l_\mu$ is the system size in the direction $\mu = x, y$. For the system shown in Fig. 2, Rashba SOI generates a spin voltage with spin direction $y$ along the $x$ axis. Up and down spins feel opposite voltage biases, hence total the spin voltage bias is

$$V^y_{1(2)} = V^\uparrow_{1(2)} - V^\downarrow_{1(2)} = \pm\partial_t\alpha_R(t)L_{12}\,, \tag{40}$$

with $L_{ij}$ the distance between lead $i$ and lead $j$ According to Eq. (38), this spin voltage drives a spin current

$$I^y_1(t) = (2\frac{e^2}{h}N_1 - G_{11} + G_{12})\,\partial_t\alpha_R(t)L_{12}\,. \tag{41}$$

To summarize, we expressed the spin current in terms of the DC charge conductance and time-dependent $\alpha_R$ in the small-$\Omega$ limit. This is our first prediction for the spin current in the low-frequency regime.

## 4.2 AC spin current in the general frequency regime

In a system driven by a time-dependent voltage, both particle and displacement currents will appear. In order to take these into account, we no longer neglect the frequency-dependence of the AC conductance, which in Fourier space is given by the admittance $G_{ij}(\omega)$. In linear response we have

$$G_{ij}(\omega) = \langle\delta I_i(\omega)/\delta V_j(\omega)\rangle\,. \tag{42}$$

In the presence of a bias voltage $V(t) = V(\omega)e^{i\omega t}$, the current $I_i(\omega)$ is calculated via the Floquet formalism. Using linear response theory, the AC conductance can then be obtained as [45–47]

$$G_{ij}(\omega) = \frac{e^2}{h}\int dE \quad Tr\{\delta_{ij}\mathbf{1}_i - S^\dagger_{ij}(E)S_{ij}(E+\hbar\omega)\}\frac{f_j(E) - f_j(E+\hbar\omega)}{\hbar\omega}\,. \tag{43}$$

The AC charge current driven by $V(\omega)$ is then given by

$$I_i(\omega) = \sum_j G_{ij}(\omega)V_j(\omega)\,. \tag{44}$$

In our case, a time-dependent SOI $\alpha_R(t) = \alpha_R(\omega)e^{i\omega t}$ leads to a (spin) voltage in the frequency domain:

$$V^y(\omega) = \pm i\omega\alpha_R(\omega)L_{12}\,, \tag{45}$$

which in turn drives a spin current

$$I^y_1(\omega) = \left(2\frac{e^2}{h}N_1 - G(\omega)_{11} + G(\omega)_{12}\right)i\omega\alpha_R(\omega)L_{12}\,. \tag{46}$$

Equation (46) is our main result for the spin (generation) conductance. It generalizes Eq. (41) to high frequencies and applies beyond the range of validity of the adiabatic approximation (see Fig. 5).

For general time-dependent $\alpha_R$, we can obtain the time-dependent spin current via the inverse Fourier transform of $I(\omega)$:

$$I^y_1(t) = \int \frac{d\omega}{2\pi}e^{i\omega t}I^y_1(\omega)\,. \tag{47}$$

## 4.3 AC spin current generation: comparison with numerics

We now perform numerical simulations to check the predictions summarized in Eqs. (41) and (46) for the spin current. The Rashba SOI has an AC-form $\alpha_R(t) = k_{so} \sin(\Omega t)$, with $k_{so} \sim 1/L$. The system size $L$ is taken to be $L = 50a$, with $a$ the lattice constant of our discretized model (see Appendix A). The precise system shape is depicted in the top right inset of Fig. 3. The width of the leads is $10a$. We choose parameters appropriate for an InAs 2DEG material with effective mass $m = 0.023 \, m_0$, and we fix the lattice spacing at $a = 2$ nm. The magnitude of the Rashba coupling is $0.8 \cdot 10^{-11}$ eVm which is in the experimental range of InAs systems [6].

We calculate the AC spin current polarized in the $y$ direction, $I^y(t) = I^y \cos(\Omega t)$, induced by the time-dependent Rashba SOI $\alpha_R(t) = k_{so} \sin(\Omega t)$. The spin current given in Eq. (35) is computed using the Floquet scattering matrix, and then compared with our analytical results in Eqs. (41) and (46).

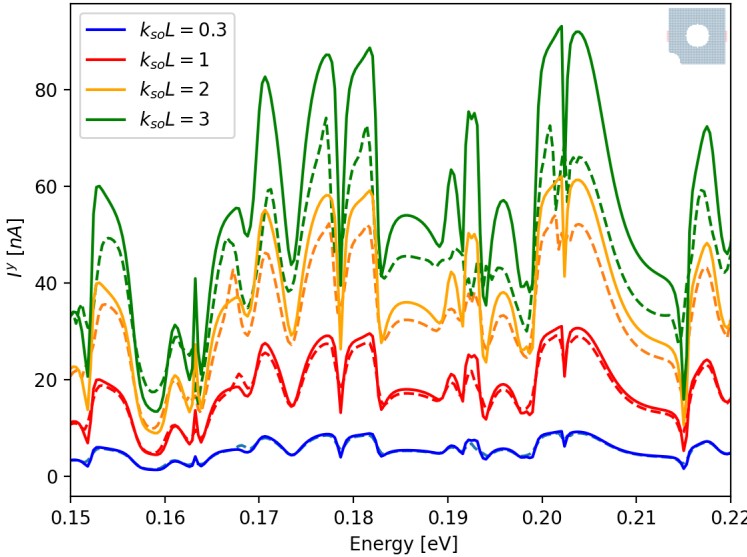

Figure 3: Comparison of the AC spin currents generated by the spin-dependent voltage (Eq. (41),solid line) and numerically (dashed) directly for the time-dependent Rashba SOI $\alpha_R(t) = k_{so} \sin(\Omega t)$ with $\Omega\tau \approx 0.3$ and varying values $k_{so}L = 0.3$ (blue), 1 (red), 2 (orange), 3 (green). The AC spin currents are plotted as a function of the Fermi energy.

We first check the validity of our approximation (41) in the low-frequency regime. We choose $\Omega/2\pi \approx 100$ GHz, corresponding to $\Omega\tau \approx 0.3$. In Fig. 3 we perform the comparison for varying values of $k_{so}$: $k_{so}L = 0.3$, 1, 2 and 3. We deliberately show a small range of the energy values to make the differences clear. Even though our approximation, in principle, requires $k_{so}L \ll \pi$, we see that up to $k_{so}L = 1$ our theory still gives quantitative agreement with the full numerical results as can be seen in Fig. 3 (red curve).

We also explore the validity of our analytical predictions as we vary $\Omega$ relative to $1/\tau$, the inverse dwell time. As outlined in App. E, we compute the dwell time using the Wigner-Smith time-delay matrix for particular Fermi energies which are determined in the range of 0.15 and 0.27 $eV$ when only 2 transverse channels are open. Taking the energy average of the dwell time gives as an approximate time of flight $\tau$ for 2 open channels.

We compare our analytical prediction with the numerical Floquet results (34) and (35) for a range of frequencies. In the low-frequency regime where $\Omega\tau < 1$, we can neglect the frequency dependence of the conductance in Eq. (43). We then use Eq. (41) to obtain our

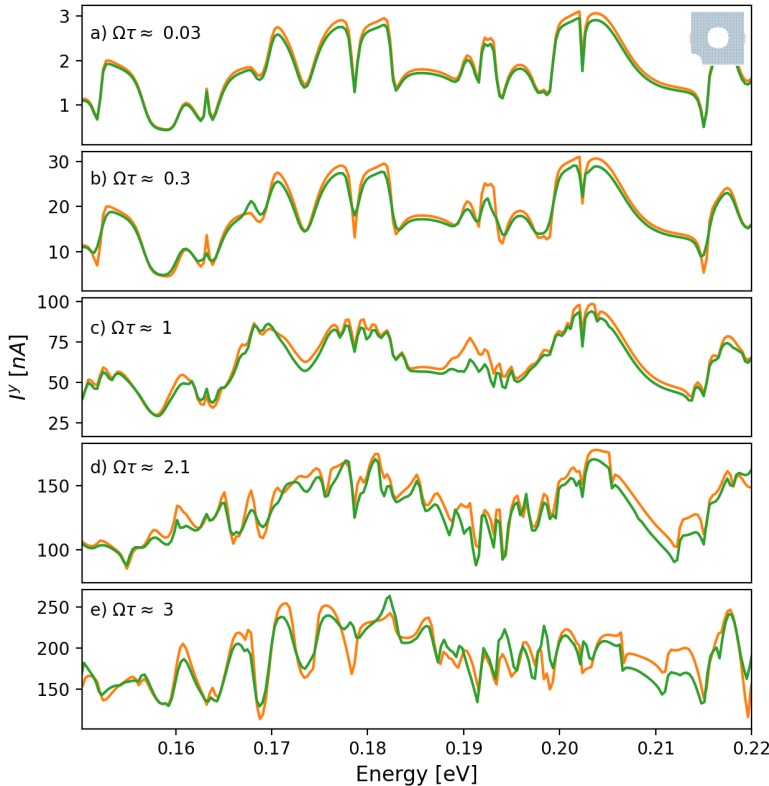

Figure 4: AC spin currents generated by the time-dependent Rashba SOI in the absence of any charge bias are plotted as a function of the Fermi energy. The numerical Floquet results (green) are compared to the spin current in $y$ direction in the presence of the time-dependent spin voltage (orange). The spin currents (orange) are calculated using Eq. (41) in the low-frequency regime in panels a) and b) and by Eq. (47) in the high-frequency regime in panels c), d) and e). The system size is $L = 50a$ and the Rashba SOI is $\alpha_R(t) = k_{so} \sin(\Omega t)$ with $k_{so}L = 1$. The frequency values are indicated at the top left corner of each panel, $\Omega\tau \approx 0.03, 0.3, 1, 2.1$ and $3$ in panels a) and b), etc., respectively. The system geometry is shown in the top right inset.

prediction and show that the result agrees well with the Floquet result as shown in Fig. 4 in panels a) and b). We note that the spin current is in the nano-Ampere range where the corresponding frequencies $\Omega/2\pi$ are approximately between 10 and 100 GHz.

In the high-frequency regime where $\Omega\tau > 1$, the AC spin current is calculated by incorporating the effect of the AC bias voltage and the admittance in Eq. (43). That is, we use Eqs. (46) and (47). We find reasonable agreement for the higher frequencies up to $\Omega\tau \approx 3$, see Fig. 4. The frequencies $\Omega/2\pi$ corresponding to those in panels c),d) and e) of Fig. 4 are 330, 700 GHz and 1 THz, respectively. These are in experimental reach. Notably, we find that in this high-frequency regime we can generate spin currents up to 250 nano-Amperes using a geometry with a length scale of about 100 nm.

We also note that the DC charge current is generated by a time-dependent Rashba SOI in left-right asymmetric systems [48, 49] beyond the adiabatic regime. As a result of having a single time-dependent parameter in our system, DC current is proportional to the square of the driving frequency [41, 48]. Hence DC current is small compared to the AC spin current.



### 4.4 Adiabatic approximation versus analytical results

In this section, we compare our analytical results with the adiabatic approximation of the Floquet scattering matrix in Eq. (37), which is obtained by the frozen scattering matrix. In Fig. 5, we choose the Rashba coupling constant $k_{so}L = 0.01$ to obtain better agreement for our analytical result. The driving frequency is chosen in the high-frequency regime, i.e., $\Omega\tau \approx 1.5$ corresponding to a driving frequency $\Omega/2\pi$ of approximately 500 GHz.

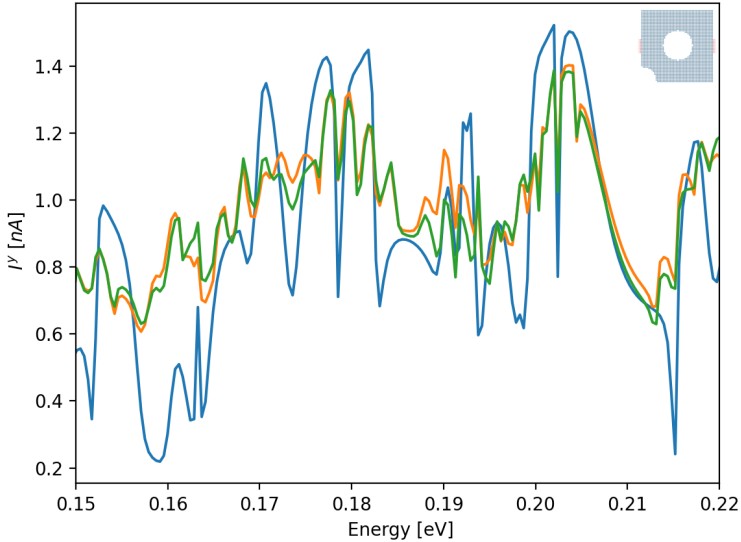

Figure 5: AC spin currents generated by the time-dependent Rashba SOI in the absence of a bias voltage are plotted as a function of the Fermi energy. In the high-frequency regime, the numerical Floquet results (green) are compared to the adiabatic approximation (blue) and the spin current in $y$-direction calculated from the time-dependent spin voltage, Eq. (47) (orange). The system size is $L = 50a$, and the Rashba SOI is $\alpha_R(t) = k_{so}\sin(\Omega t)$ with $k_{so}L = 0.01$ and $\Omega\tau \approx 1.5$.

The adiabatic approximation should fail beyond frequencies around $\Omega\tau \approx 1$. Indeed, Fig. 5 shows that the adiabatic approximation breaks down at high frequencies while our prediction (47) still shows good agreement with the numerical Floquet result. We note that in addition to having a much wider range of validity, the computation of our analytical result is much easier and takes much less time compared to the frozen scattering matrix approximation.

## 5 Charge signal from a spin current

### 5.1 Dynamical SOI-based spin transistor setting

In this section, we discuss how to detect the AC spin current generated by the spin potential induced through a time-dependent SOI. Usually this can be achieved with a device with nonzero spin (detection) conductance. While the conventional means of detecting spin currents are based on ferromagnetic leads, here we will exploit the SU(2) gauge structure of the Rashba SOI further, to design a device with a non-zero spin magnetic field. The latter can then convert spin currents into charge signals [16].

The spin (detection) conductance in Eq. (28) is converted into the difference of two charge magneto-conductances according to Eq. (31). Then an Onsager relation implies that the total charge conductance for a 2-lead system vanishes in the presence of time-reversal symmetry.

On the other hand, either breaking the time-reversal symmetry e.g. via an applied magnetic field or connecting a third terminal to the system [17] results in a nonzero charge conductance and an AC charge current. As a result, controlling the symmetry properties of the system, we can obtain on/off states of this dynamical spin transistor (see Fig. 8). Below we will take the route where we connect a third terminal.

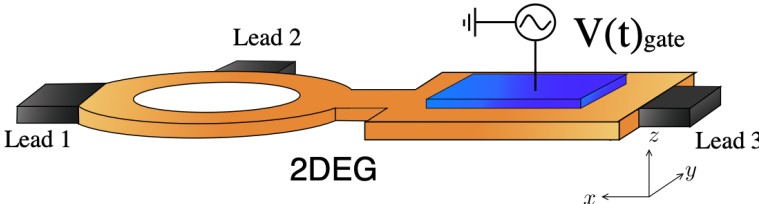

Figure 6: Setup of the dynamical spin-orbit based spin transistor with the top terminal (Lead 2) controlled electrostatically. The AC gate voltage in the right cavity induces a dynamical Rashba SOI, which creates an AC spin current that flows into the ring on the left. There, due to a spatially inhomogeneous Rashba SOI, the spin current is transformed into a nonzero charge current if the number of leads attached to the ring is greater than 2; the charge current is strongly suppressed in the two-terminal ring.

To explore how this conversion works in detail, we consider a system that consists of two subsystems, containing a spatially inhomogeneous and a time-dependent Rashba SOI, respectively, see Fig. 6. In Fig. 6, the right subsystem consists of a 2D quantum wire (with length $L$ and width $W$) with a time-dependent SOI $\alpha_R(t) = k_{so}\sin(\Omega t)$ engineered by an AC gate voltage. The left subsystem contains a ballistic ring (with diameter $L$) with a spatially inhomogeneous SOI $\alpha_R(\mathbf{x}) = k_{so}(L-y)/L$. This way spin currents generated in the right subsystem via a spin electric field are converted into charge currents via a spin magnetic field in the left subsystem.

## 5.2 AC charge signal in the low-frequency regime

In the absence of a bias voltage $V$, the charge current in lead 1 follows from Eq. (28) as

$$I_1 = -\sum_{j,b} G_{1j}^{0b} V_j^b \,, \tag{48}$$

with spin direction $b = x, y, z$. We express the spin conductance in terms of the charge magneto-conductance (31). In the low-frequency regime, we again neglect the frequency dependence of the conductance. Then the charge current in lead $i$ reads

$$I_1 \approx -\sum_b \sum_j (G_{1j}(\mathcal{B}^b) - G_{1j}(-\mathcal{B}^b)) f^b V_j^b \,. \tag{49}$$

For spatially inhomogeneous SOI $\alpha_R(\mathbf{x}) = k_{so}(L-y)/L$ we obtain from Eq. (14) the spin magnetic field components $\mathbf{B}_z^a$ in the $x$- and $y$-directions as

$$\begin{aligned} \mathcal{B}^x &= -\partial_x \alpha_R(\mathbf{x}) f^x \,, \\ \mathcal{B}^y &= -\partial_y \alpha_R(\mathbf{x}) f^y \,. \end{aligned} \tag{50}$$

The spin voltages generated by the time-dependent SOI $\alpha_R(t) = k_{so}\sin(\Omega t)$ are obtained from Eq. (12) as

$$V_{1(3)}^y = V_{1(3)}^{\uparrow} - V_{1(3)}^{\downarrow} = \pm\partial_t \alpha_R(t) L \,. \tag{51}$$

The charge current in lead 1 can now be computed as

$$I_1(t) = G_s \partial_t \alpha_R(t) L, \tag{52}$$

where

$$G_s = -G_{11}(\mathcal{B}^y) + G_{11}(-\mathcal{B}^y) + G_{13}(\mathcal{B}^y) - G_{13}(-\mathcal{B}^y). \tag{53}$$

This quantifies our prediction that the spin current can be converted into a charge current in a spatially inhomogeneous system by means of a spin magnetic field.

### 5.3 AC charge signal in the high-frequency regime

As explained in Section 4.2, at high-frequency we should retain the frequency dependence of the AC conductance. We follow the same steps as in Section 4.2 and rewrite the AC charge current expression (49), but retaining the frequency dependence given in Eq. (43), as

$$I_1(\omega) \approx -\sum_b \sum_j [G_{1j}(\omega, \mathcal{B}^b) - G_{1j}(\omega, -\mathcal{B}^b)] f^b V(\omega)_j^b, \tag{54}$$

where the spin-dependent magnetic field is $\mathcal{B}^y = k_{so} L^{-1} f^y$ and the spin voltage is $V^{\uparrow(\downarrow)}(\omega) = \pm i \omega \alpha(\omega) L$. This constitutes our main result for the spin (detection) conductance, which applies to both high- and low-frequency regimes.

### 5.4 Simulating the dynamical spin-transistor functionality

We performed numerical transport simulations of the spin transistor in Fig. 6 and explore the range of validity of our analytical result (54). We choose the time-dependent Rashba SOI to be $\alpha_R(t) = k_{so} \sin(\Omega t)$ and the spatially inhomogeneous Rashba SOI to be $\alpha_R(\mathbf{x}) = k_{so}(L-y)/L$, where the Rashba SOI constant is selected such that $k_{so}L = 1$, where $L$ is the system size along the $x$ direction for the right subsystem. The shape of the system is shown at the top right corner of Fig. 6. The right part is a 2D wire with length $L = 50a$, width $W = 30a$ and one connected lead of width $10a$. The left part is a ballistic ring with an inner radius of $10a$, an outer radius of $25a$, and two connected leads of width $10a$. The two parts are connected via a bridge of width $10a$ made of the same material.

We consider 3 open channels and choose the Fermi energy between 0.29 and 0.47 eV. After computing the dwell time to specify the range of the frequency, we calculate the AC charge currents $I(t) = I \cos(\Omega t)$ in the absence of the bias voltage on the left using the Floquet scattering matrix given in Eq. (34) and Eq. (35). Finally, we compare the Floquet result for AC charge current with our analytical prediction both in the low and the high-frequency regimes. Our analytical results are obtained from Eq. (52) for $\Omega \tau \approx 0.1, 0.3$ and shown in Fig. 6a) and b). Here, the frequencies $\Omega/2\pi$ are approximately 40 and 120 GHz. We find excellent agreement between our analytical results and the numerically obtained AC charge current. Moreover, even in the low-frequency regime, we see that the mechanism produces few-nA charge currents that are experimentally observable. For higher frequencies where $\Omega > 1/\tau$, we compare our prediction based on Eq. (54) with the numerical dynamical conductance using driving frequencies $\Omega \tau \approx 1, 1.5$, and 2.1. These values correspond to $\Omega/2\pi$ being approximately 400, 600, and 840 GHz. In Fig. 6c),d),e) we also observe on the whole a fairly good match between the two currents in the high-frequency regime up to $\Omega \tau \approx 2.1$.

In Fig. 8, we show the on/off signals of the dynamical spin transistor. The AC charge currents in these plots are computed using the Floquet scattering matrix. In particular the transistor is in the off-state if the number of leads attached to the left ring is equal to 2. Hidden Onsager relations that apply to 2-terminal devices ensure that the spin-charge conversion is suppressed. Connecting a third terminal to the system results in non-zero charge current as seen from Fig. 8: this is the spin transistor on-state.

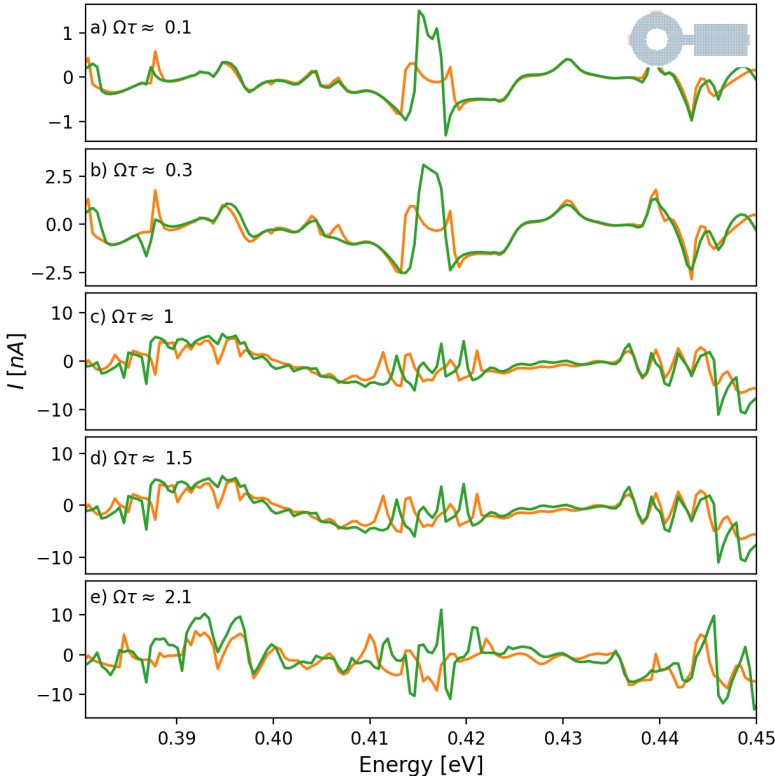

Figure 7: AC charge current in Lead 1, see Fig. 6, generated by the time-dependent and spatially inhomogeneous Rashba SOI. Numerical results based on the Floquet formalism (green) are compared to the current generated by the spin-dependent voltage and pseudo-magnetic field (orange) in the low- and high-frequency regime. The currents are plotted as a function of the Fermi energy. The analytical results in a) and b) are calculated based on Eq. (52) and in panels c) to e) using Eq. (54). The dynamical Rashba SOI in the right half system is $\alpha_R(t) = k_{so} \sin(\Omega t)$ with $k_{so}L = 1$ and the spatially inhomogeneous Rashba SOI in the left half system (the ring) is $\alpha_R(\mathbf{x}) = k_{so}(L-y)/L$. The size of each system part is $L = 50a$. The frequencies used are $\Omega\tau \approx 0.1, 0.3, 1, 1.5$ and $2.1$ from top to bottom panel.

# 6  Conclusions

We considered the generation and detection of pure spin currents in mesoscopic 2DEG cavities with time- and space-modulated Rashba SOI. Such modulations can be realized by applying local gate voltages, allowing an all-electrical architecture. Our underlying concept is based on the non-Abelian gauge structure of spin-orbit fields. By means of a tailored $SU(2)$ gauge transformation we converted the general Hamiltonians with time-periodic or spatially inhomogeneous Rashba SOI into corresponding approximate Hamiltonians involving instead emergent $U(1)$ fields.

For the time-dependent Rashba SOI, the $U(1)$ fields correspond to spin-dependent voltages with opposite associated electrical fields for spin-up and -down electrons, respectively. Then the total spin-dependent voltage becomes $V^\uparrow - V^\downarrow$ to linear order in the Rashba SOI strength. Thus the action of the time-dependent Rashba SOI can be considered as a spin electric force, that enables to generate a pure AC spin current in the absence of an applied bias voltage.

Analogously, in the presence of a spatially inhomogeneous Rashba SOI, one can show that an appropriate gauge transformation relates the Rashba spin-orbit fields to pseudo-magnetic

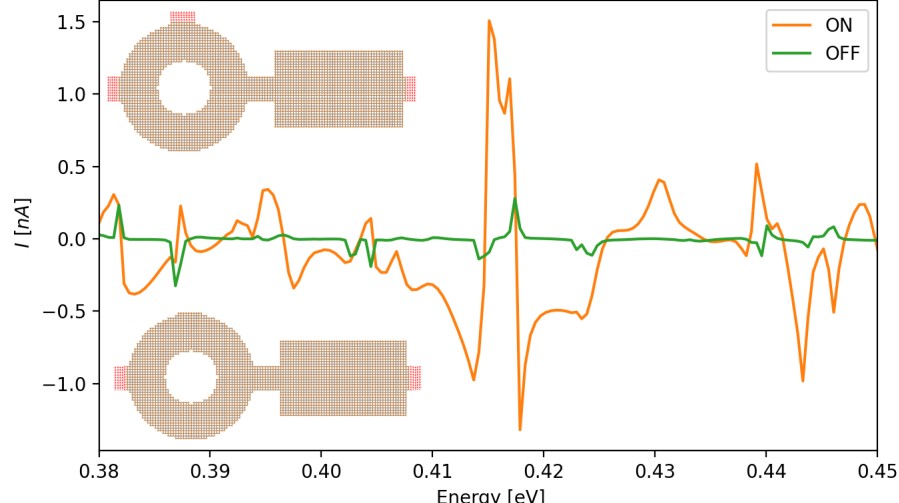

Figure 8: Demonstration of the on/off states of dynamical spin transistor. AC charge currents in Lead 1 as a function of the Fermi energy, generated by time-dependent and spatially inhomogeneous Rashba SOI, is plotted. Numerical results are based on the Floquet formalism. The charge current is calculated for both 2-Lead (inset below) and 3-Lead (inset above) systems. The dynamical Rashba SOI in the right half system is $\alpha_R(t) = k_{so}\sin(\Omega t)$ with $k_{so}L = 1$ and the spatially inhomogeneous Rashba SOI in the left half system (the ring) is $\alpha_R(\mathbf{x}) = k_{so}(L - y)/L$. The size of each system part is $L = 50a$ and the driving frequency is $\Omega\tau \approx 0.1$.

fields $\mathcal{B}$ with opposite directions for the two spin species. Thereby, the spin conductance is rewritten as a charge magneto-conductance where opposite spins experience anti-parallel spin magnetic fields: $G^{\uparrow} - G^{\downarrow} \approx G(\mathcal{B}) - G(-\mathcal{B})$. Thereby the generated AC spin current due to the dynamical Rashba coupling can be read out as an AC charge current by means of an appropriately devised spatially inhomogeneous SOI. Moreover, in view of an Onsager relation and current conservation, the AC-generated conductance difference, $G(\mathcal{B}) - G(-\mathcal{B})$, is suppressed for a two-terminal setting in presence of time-reversal symmetry. Therefore the transistor on/off state can be controlled via the coupling to a third lead or upon changing the symmetry properties of the system. By combining all these aspects, we devised and proposed a dynamical SOI-based spin-transistor which integrates a controllable spin current generation and its detection solely by electric means.

We checked our analytical approximations by means of numerical calculations using Floquet theory which allows us to calculate time-periodic spin and charge currents within a Floquet scattering matrix formalism. We presented results and compared the different approaches both in the regimes of adiabatic ($\Omega\tau \ll 1$) and high-frequency ($\Omega\tau \geq 1$) driving. The major assumptions behind our analytical derivations are $k_{so}L \leq \pi$ and $(k_{so}L)(\hbar\Omega) \leq \epsilon_F$.

The latter bound implies that our approach is valid and facilitates quantum calculations in a frequency regime that extends far beyond the adiabatic limit, often assumed in analytical treatments.

The driving frequencies considered cover a broad window ranging, after rescaling, up to frequencies of about 1 THz. Our numerical simulations show furthermore that an AC spin current of several hundred nano-Amperes can be obtained in a 100 nm 2DEG InAs system for driving frequencies of about 1 THz and a realistic Rashba coupling $0.8 \cdot 10^{-11}$ eVm. This shows that an experimental realization of the dynamic spin transistor functionality proposed is in experimental reach.

## Acknowledgements

We thank M. Wimmer and A.M. Bozkurt for helpful discussions. IA and FNG are grateful to the hospitality of Regensburg University where parts of this project were carried out. FNG is also grateful to the hospitality of Utrecht University.

**Funding information:** This work was supported by TUBITAK under grant 110T841 and the programme 2214-A, and by the Deutsche Forschungsgemeinschaft (German Science Foundation), Project-ID 314695032-SFB 1277 (project A07).

## A   Tight-binding Hamiltonian for 2DEG with Rashba SOI

In the case of a 2DEG with spatially inhomogeneous Rashba SOI in the $x$-$y$-plane, the continuum Hamiltonian we use for our analytical calculations reads

$$H = \frac{\hbar^2}{2m^*}(k_x^2 + k_y^2) + \frac{1}{2}\left\{\alpha_R(\mathbf{x}), \left(\sigma^x p_y - \sigma^y p_x\right)\right\}. \tag{A.55}$$

where $\alpha_R(\mathbf{x})$ is the Rashba SOI strength, $m^*$ is effective mass of an electron and $\sigma$ denotes Pauli matrices. In our numerical simulations, we use the discretized version of the Hamiltonian (A.55) [50,51]. It is defined on a 2D square lattice with a lattice constant $a$,

$$
\begin{aligned}
H_{tb} = &\sum_{k,l,\sigma,\sigma'} 4t\,(c^\dagger_{k,l,\sigma}c_{k,l,\sigma'}) + \sum_{k,l,\sigma} t\,(c^\dagger_{k+1,l,\sigma}c_{k,l,\sigma} + c^\dagger_{k,l+1,\sigma}c_{k,l,\sigma}) + \\
&\sum_{k,l,\sigma,\sigma'} \frac{1}{2a}\frac{1}{2}(\alpha_{R,k} + \alpha_{R,k+1})\,c^\dagger_{k+1,l,\sigma}(i\sigma_y)^{\sigma\sigma'}c_{k,l,\sigma'} + \\
&\sum_{k,l,\sigma,\sigma'} -\frac{1}{2a}\frac{1}{2}(\alpha_{R,l} + \alpha_{R,l+1})\,c^\dagger_{k,l+1,\sigma}(i\sigma_x)^{\sigma\sigma'}c_{k,l,\sigma'}, \tag{A.56}
\end{aligned}
$$

where $c^\dagger_{k,l,\sigma}$ is the operator that creates an electron with spin $\sigma$ at the lattice point $(k,l)$, and the hopping amplitude is $t = -\hbar^2/(2m^*a^2)$.

## B   Floquet Hamiltonian with time-dependent Rashba SOI

We review the conversion of a time-dependent Hamiltonian into a static matrix Hamiltonian with Floquet states [52]. Floquet theory is particularly functional for understanding the behavior of quantum mechanical systems with a time-periodic Hamiltonian:

$$H(\vec{r}, t + \mathcal{T}) = H(\vec{r}, t),$$

with period $\mathcal{T} = 2\pi/\Omega$ and $\Omega$ is given by the frequency of the periodically driven potential. The solutions of the time-dependent Schrödinger equation are given by the Floquet wave functions.

Consider a 2D quantum mechanical system with a time-periodic Hamiltonian that can be separated into a static and a time-dependent part as

$$H(\vec{r}, t) = H_0(\vec{r}) + H_1(\vec{r}, t), \tag{B.1}$$

where the kinetic term is $H_0(\vec{r}) = -\frac{\hbar^2}{2m}(\partial_x^2 + \partial_y^2)$. The time-dependent Schrödinger equation can be put into the form

$$H_F(\vec{r}, t)|\Psi(\vec{r}, t)\rangle = 0, \tag{B.2}$$

where the hermitian Floquet Hamiltonian $H_F(\vec{r}, t)$ is related to the original Hamiltonian through

$$H_F(\vec{r}, t) = H(\vec{r}, t) - i\hbar \frac{\partial}{\partial t} . \qquad \text{(B.3)}$$

The solution of Eq. (B.2) is formally given by the Floquet states

$$|\Psi_\eta(\vec{r}, t)\rangle = e^{-iE_\eta t/\hbar}|\phi_\eta(\vec{r}, t)\rangle , \qquad \text{(B.4)}$$

where $|\phi_\eta(\vec{r}, t)\rangle$ is called a Floquet mode. Floquet modes are eigenfunctions of the Floquet Hamiltonian $H_F(\vec{r}, t)$ with eigenvalues (quasi-energies) $E_\eta$:

$$\left(H(t) - i\hbar \frac{\partial}{\partial t}\right)|\phi_\eta(\vec{r}, t)\rangle = H_F(\vec{r}, t)|\phi_\eta(\vec{r}, t)\rangle = E_\eta|\phi_\eta(\vec{r}, t)\rangle . \qquad \text{(B.5)}$$

with periodic eigenfunctions

$$|\phi_\eta(\vec{r}, t)\rangle = \sum_n e^{-in\Omega t}|n\rangle, \quad |\phi_\eta(\vec{r}, t)\rangle = |\phi_\eta(\vec{r}, t + \mathcal{T})\rangle . \qquad \text{(B.6)}$$

Here $|n\rangle$ are basis vectors of the eigenfunctions of the Floquet Hamiltonian and refer to the Floquet states. In Fourier space, the time-dependent Hamiltonian can now be expressed as a static matrix Hamiltonian

$$H_F = \sum_{n=-\infty}^{\infty} \left( (H_0 - n\hbar\Omega)|n\rangle\langle n| + \frac{iH_1}{2}(|n\rangle\langle n+1| - |n\rangle\langle n-1|) \right) . \qquad \text{(B.7)}$$

Hence the original time-periodic system has been converted into a multi-channel stationary system expanded in the Floquet states. Using the Hamiltonian (B.7), we can numerically compute the Floquet states for each Floquet channel to obtain the elements of the Floquet scattering matrix.

In the case of the Rashba SOI $\alpha(t) = k_{so}\sin(\Omega t)$ we have $H_1 = ik_{so}(\sigma_y \partial_x - \sigma_x \partial_y)$.

## C Comparing the Floquet scattering matrix with its adiabatic approximation

For a further analysis of our spin current calculations, we compare results based on the full Floquet scattering matrix $S_F(E_n, E)$ with that of the adiabatic approximation Eq. (37) for different frequencies. This allows us to check the range of validity of the adiabatic approximation. In Fig. 9, we compare the adiabatic and full Floquet spin currents for the chaotic ballistic system of Fig. 2 for low and high frequencies. The results agree in the adiabatic limit $\Omega\tau \approx 0.1$ as expected and show semi-quantitative agreement up to the frequency of $\Omega\tau \approx 1$.

## D Spin and charge current calculation in the Landauer-Büttiker Formalism

Here we sketch the derivation of the expressions (34) and (35) for the AC spin/charge current using the Landauer-Büttiker formalism. We start with the charge/spin current operator. Recall that $\alpha = 0, x, y, z$ indicates the components of the charge current and the spin current, respectively:

$$\hat{I}_i^\alpha(t) = \frac{e}{4\pi} \int dE dE' \, e^{i(E-E')/\hbar}[\hat{a}_{im,\sigma}^\dagger(E)\sigma^\alpha \hat{a}_{im,\sigma'}(E') - \hat{b}_{im,\sigma}^\dagger(E)\sigma^\alpha \hat{b}_{im,\sigma'}(E')], \qquad \text{(D.1)}$$

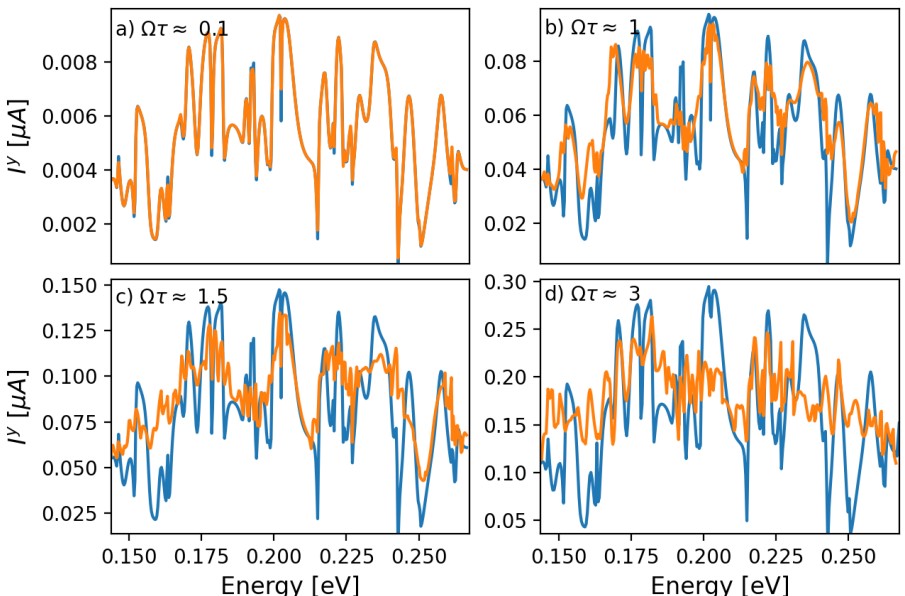

Figure 9: Spin currents for the chaotic ballistic system, Fig. 2, calculated by using the full Floquet scattering matrix (orange) and by the adiabatic approximation, Eq. (37) (blue) for $k_{so}L = 1$ and $\Omega\tau \approx 0.1$ (a), 1 (b), 1.5 (c) and 3 (d).

where $\hat{a}^\dagger$ ($\hat{a}$) and $\hat{b}^\dagger$ ($\hat{b}$) are the creation (annihilation) operators for incident and scattered electrons, respectively, $m$ denotes the channel in lead $i$, and $\sigma$ represents the spin operator. Note that Eq. (D.1) is derived under the assumption $E - E' \ll \epsilon_F$.

For a periodically driven system, the relation between the operators $\hat{a}$ and $\hat{b}$ is given by the Floquet scattering matrix

$$\hat{b}_{im,\sigma}(E) = \sum_{n=-\infty}^{\infty} \sum_{j=1}^{N_r} \sum_{m'\in j} \sum_{\sigma'=\pm 1} S_{F,im,jm'}^{\sigma\sigma'}(E, E_n)\hat{a}_{jm',\sigma'}(E_n), \qquad (D.2)$$

where $n$ denotes the Floquet states with $E_n = E + n\hbar\Omega$. We write the expectation value of the current as inverse Fourier series,

$$I_i^\alpha(t) \equiv \langle\hat{I}_i^\alpha(t)\rangle = \sum_{l=-\infty}^{\infty} e^{-il\Omega t} I_{i,l}^\alpha, \qquad (D.3)$$

with $I_{i,l}^\alpha = \langle\hat{I}_i^\alpha(l\Omega)\rangle$. This yields [32], in view of Eqs. (D.2) and (D.5),

$$I_{i,l}^\alpha = \frac{e}{4\pi} \int_{-\infty}^{\infty} dE \sum_{n=-\infty}^{\infty} \sum_{i=1}^{N_r} \sum_{m\in i, m'\in j} \{f_j(E) - f_i(E_n)\} Tr[S_{F,im,jm'}^\dagger(E_n, E)\sigma^\alpha S_{F,im,jm'}(E_{l+n}, E)]. \quad (D.4)$$

Here we use that the quantum statistical average of the operator $\langle\hat{a}^\dagger\hat{a}\rangle$ gives the Fermi function for electrons in lead $i$,

$$\langle\hat{a}_{im,\sigma}^\dagger(E)\hat{a}_{jm',\sigma'}(E')\rangle = \delta_{ij}\delta_{mm'}\delta_{\sigma\sigma'}\delta(E - E')f_i(E). \qquad (D.5)$$

When the chemical potential difference between the leads vanishes, the Fermi functions are equal, $f_i(E) = f_j(E) = f_0(E)$. Furthermore in the limit $\Omega \to 0$ one can expand the Fermi

function with energy $E_n = E + n\hbar\Omega$ and obtains

$$f_0(E) - f_0(E_n) = -\frac{df_0(E)}{dE}n\hbar\Omega + \mathcal{O}(\Omega^2). \tag{D.6}$$

Substituting into Eq. (D.4), we obtain

$$I_{i,l}^\alpha = \frac{e}{4\pi}\int_{-\infty}^{\infty}dE\left(-\frac{\partial f_0}{\partial E}\right)\sum_{n=-\infty}^{\infty}\sum_{i=1}^{N_r}\sum_{m\in i,m'\in j}(n\hbar\Omega)Tr[S_{F,im,jm'}^\dagger(E_n,E)\sigma^\alpha S_{F,im,jm'}(E_{l+n},E)]. \tag{D.7}$$

In the zero-temperature limit, only electrons at the Fermi energy contribute to the current $[\frac{\partial f_0}{\partial E} = \delta(E - E_F))]$, and the integral can be evaluated. This yields

$$I_{i,l}^\alpha = \frac{e}{4\pi}\hbar\Omega\sum_{n=-\infty}^{\infty}\sum_{j=1}^{N_r}\sum_{m\in i,m'\in j}nTr[S_{F,imjm'}^\dagger(E_{F,n},E_F)\sigma^\alpha S_{F,im,jm'}(E_{F,l+n},E_F)]. \tag{D.8}$$

We calculate the full time-dependent current by substituting this expression into Eq. (D.3).

# E  Calculation of the dwell time

Here we calculate the dwell time of an electron spent in the scattering system with a time-dependent potential. The dwell time is given by the energy derivative of the scattering phase shift in terms of scattering matrix [43, 44]. The diagonal elements of the Wigner-Smith time-delay matrix,

$$Q(E) = -i\hbar S^\dagger(E)\frac{\partial S(E)}{\partial E}, \tag{E.1}$$

are real and give the proper time delays for each transport channel. Taking the average overall channels, one obtains the time-delay

$$\tau_W(E) = \frac{1}{N}Tr\{Q\}. \tag{E.2}$$

For the time-periodic systems, we compute the dwell time using the Floquet scattering matrix,

$$\tau_W(E_F) = -\frac{i\hbar}{N'}Tr\left\{\sum_n S_F^\dagger(E_F,E_{F,n})\frac{\partial S_F(E_F,E_{F,n})}{\partial E}\right\}, \tag{E.3}$$

where $N' = 2N(2n+1)$ is the total number of scattering channels that results from the total number of Floquet bands $n$ and spin polarization. We also compute the energy averaged dwell time denoted as $\tau$. The energy window is chosen such that the number of open channels is constant. This averaged dwell time $\tau$ is a convenient measure to physically distinguish the low-frequency and high-frequency regimes $\Omega < 1/\tau$ and $\Omega > 1/\tau$, respectively.

# F  Role of the number of Floquet bands for the numerical results

Here we present our procedure to determine the number of the necessary Floquet channel for performing numerically converged simulations. In the calculation of the spin current using Eq. (32) we need to choose a cut-off value $n_{max}$ in the sum over the Floquet side bands.

The ratio between the amplitude of the oscillating potential and the driving frequency determines the cut-off value $n_{max}$. As discussed in Ref. [32](see chapter 3), if the amplitude of

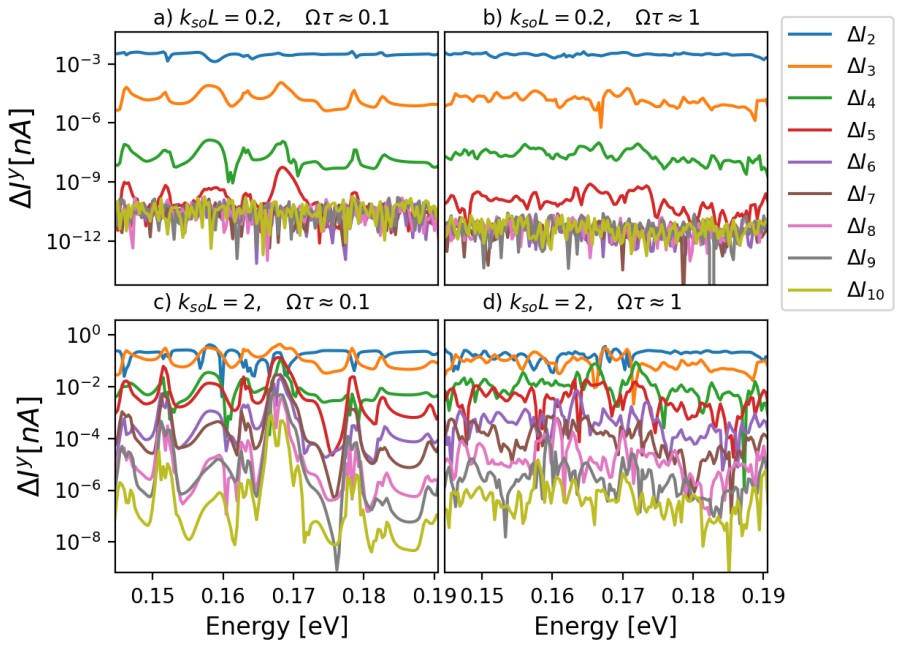

Figure 10: The convergence in $n$ of the AC spin currents generated by the time-dependent Rashba SOI, where $\Delta I_n = (I_n - I_{n-1})/I_n$ and $I_n$ is the current calculated with maximum Floquet states $N$. The parameters correspond to $\Omega\tau \approx 0.1$ and $1$, and $k_{so}L = 0.2$ and $2$.

the oscillating potential is much smaller than $\hbar\Omega$, $n_{max} = 1$ is sufficient for numerical calculations. On the other hand, if the amplitude of the oscillating potential is comparable to the driving frequency, one needs $n_{max} > 1$ for numerical calculations.

It is essential to know the number of the bands prior to numerical calculations as the time cost of a calculation heavily depends on this number. To determine this number, we compute the spin current for a selected number of Floquet states $2n + 1$ and change $n$ to calculate the precision defined by $\Delta I_n = (I_n - I_{n-1})/I_n$. This quantity is plotted in Fig.10 for the system in section 4.3 with the parameters $k_{so}L = 0.2$ and $2$ and for $\Omega\tau \approx 0.1$ and $1$. Generally, we observe that smaller cut-off values $n_{max}$ suffice for smaller choices of $k_{so}$, independent of frequency, as expected. At moderate values of $k_{so}$, while smaller $n_{max}$ suffice for smaller frequencies, higher $n_{max}$ is needed at higher frequencies. We find that our chosen cut-off value $n_{max} = 10$ provides a sufficiently good approximation to the time-dependent currents for the range of parameters considered in this Paper.

# G Generation of the higher harmonics of spin and charge current

We also investigate higher harmonics generation of spin and charge currents by time-dependent Rashba SOI. We consider the chaotic ballistic system connected to two leads from Sec. 4.3 with the parameters Rashba SOI $\alpha(t) = k_{so}\sin(\Omega t)$, $k_{so}L = 1$ and $\Omega\tau \approx 1$, and calculate high harmonic spin and charge currents up to $l = 5$ using the Eqs. 34 and 35. We find that the induced spin current polarized in the x and y directions feature odd harmonics. In contrast, the induced spin current polarized in the z direction as well as the induced charge current receive contributions from the even harmonics. This can be seen from the symmetry property

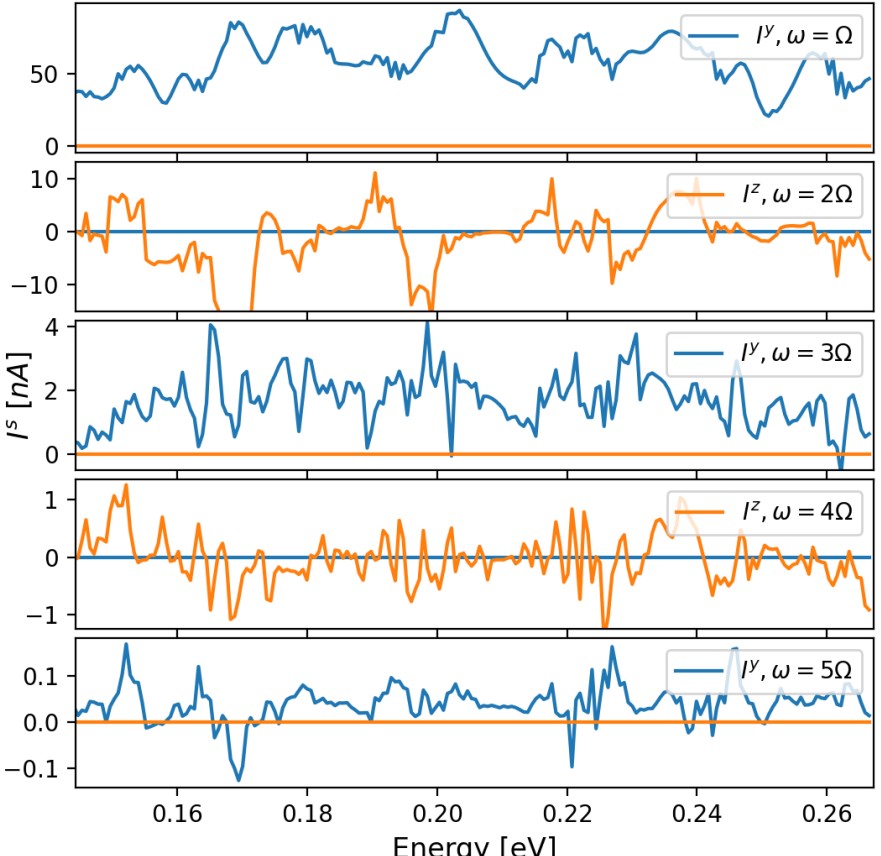

Figure 11: Higher harmonics of the induced spin currents in y (blue) and z (orange) direction in the presence of a time dependent Rashba SOI ($\alpha(t) = k_{so} \sin(\Omega t)$). The currents are plotted as a function of the Fermi energy for $k_{so} = 1/L$ and , where $L$ is the system size in the x-direction. The frequency is chosen such that $\Omega \tau \approx 1$, where $\tau$ is the time of flight.

of a generic Rashba 2DEG Hamiltonian

$$\sigma_z H(\alpha) \sigma_z = H(-\alpha). \tag{G.1}$$

Since the $x$- and $y$-polarised spin currents are odd under the same transformation ($\sigma_z I^{x,y} \sigma_z = -I^{x,y}$), their expectation values will be odd functions of $\alpha$. In contrast, the charge current and the spin current polarized in the $z$-direction is even ($\sigma_z I^{z,c} \sigma_z = I^{z,c}$), hence their expectation values will be even functions of $\alpha$. Our simulations agree with this: Fig. 11 shows the $y$- and $z$-polarised spin current as a function of the Fermi energy. Note that the first harmonic term is an order of magnitude higher than the higher harmonics contributions.

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
