# Peer review of "Dynamical Spin-Orbit-Based Spin Transistor"

_SciPost Physics, doi:SciPost Phys. 14, 060 (2023)_

## Round 1 · Referee Report · Anonymous · 2022-2-8

Strengths
The manuscript meets "4. Provide a novel and synergetic link between different research areas." acceptance criterion. See also my Report.
Weaknesses
See my Report.
Report
The manuscript combines two decades of studies in spintronics, focused on 2D electron systems with the Rashba spin-orbit coupling, with pumping by time-dependent potentials (in the manuscript case, this is time-dependent Rashba spin-orbit coupling controlled by AC gate voltage) to propose interesting new type of spin transistor. This is interesting result, and in my opinion manuscript satifies "4. Provide a novel and synergetic link between different research areas." as the criterion of acceptance in SciPost Physics.
However, I see the need to improve current presentation as follows:
1. Spin current generation by time-dependent Rashba coupling has already been discussed long ago in cited Ref. 8. Thus, a paragraph is needed to describe how the present manuscript offers something new in terms of physics and theoretical treatment. For example, the authors usage of Floquet scattering matrix for this problem is certainly far more sophisticated than what was done in Ref. 8, and this methodology is also more general than adiabatic scattering matrix approach (also employed in the manuscript, with limitations clearly explained) which is the dominant approach for other types of spin pumping problems in spintronics,
see https://journals.aps.org/rmp/abstract/10.1103/RevModPhys.77.1375.
2. Equation 34 is written without any citation, which gives impression that the authors have derived it in the present manuscript. However, this seems to be just usage of Equations from Sec. 3.3.1 of Moskalets' book, cited as Ref. 23, so the authors should add appropriate citations here and in Appendices.
3. The authors evalute Eq. 34 using just one harmonic. However, very recent studies:
A. High-harmonic generation in spin-orbit coupled systems
Phys. Rev. B 102, 081121(R) – Published 26 August 2020
B. High-harmonic generation in spin and charge current pumping at ferromagnetic or antiferromagnetic resonance in the presence of spin-orbit coupling
https://arxiv.org/abs/2112.14685
point out that time-dependent Rashba systems invariably generate higher than one harmonic in pumped spin and/or charge currents. Thus, this is my MAIN suggestion which could CONNECT paper with this contemporry direction -> try to compute higher harmonics in pumped current. Also, as discussed in B. above, once the device is left-right asymmetric [see also Fig. 4 in J. Phys.: Mater. 2, 025004 (2019) or Phys. Rev. B 72, 245339 (2005) for detailed discussion about symmetry breaking requirements], one should observe DC charge current without any second "2DEG converted" in Fig. 1. I am wondering if the authors have checked that by simply computing Eq. 34 for charge current.
4. The definition of adiabatic limit, "If the frequency is much smaller than the inverse of the time of
fight, ..., the scattering process is considered to be in the adiabatic limit.", does not take into account the relationship between frequency and Fermi energy. That is, one can see immediately from Eq. 34 that integration is not needed, which is adiabatic, if frequency is much smaller than the Fermi energy. Some comment on the connection between these definitions would be useful.
5. In Fig. 9 it will look puzzling to experts in general theory of Floquet systems that the results are NOT very sensitive to the driving frequency (left versus right panels). Usually it is the combination of both the driving frequency and amplitude which sets the regime.
6. The authors repeatedly use term "spin-motive force", including in abstract and conclusion:
"For the time-dependent Rashba SOI, the U(1) elds correspond to spin-dependent voltages
with opposite associated electrical elds for spin-up and -down electrons, respectively. Then
the total spin-dependent voltage becomes Vup-Vdown to linear order in the Rashba SOI strehgth.
Thus the action of the time-dependent Rashba SOI can be considered as a spin-motive force,
that enables to generate a pure AC spin current in the absence of an applied bias voltage."
However, this term is widely accepted to denote CHARGE pumping by dynamical noncollinear and noncoplanar magnetic textures (such as domain walls), see, e.g., theory of:
Equation-of-motion approach of spin-motive force
Journal of Applied Physics 109, 07C735 (2011); https://doi.org/10.1063/1.3565398
and references therein, or experiments such as:
Time-Domain Observation of the Spin-motive Force in Permalloy Nanowires
Phys. Rev. Lett. 108, 147202 – Published 5 April 2012
So, to avoid confusion, they should change this terminology.
7. Current manuscript has many typos: under Eq. 52 "prediction demostrating"; "Figure 7: Dyanmical spin transistor", ... Please run spell checker.
Requested changes
See my Report.
Author: Fahriye Nur Gursoy on 2022-10-06 [id 2891]
(in reply to Report 1 on 2022-02-08)
We thank the referee for their thoughtful report and for recommending that our work be published in Scipost physics. The referee also requested certain changes, and we respond item by item below.
- Spin current generation by time-dependent Rashba coupling has already been discussed long ago in cited Ref. 8. Thus, a paragraph is needed to describe how the present manuscript offers something new in terms of physics and theoretical treatment. For example, the authors usage of Floquet scattering matrix for this problem is certainly far more sophisticated than what was done in Ref. 8, and this methodology is also more general than adiabatic scattering matrix approach (also employed in the manuscript, with limitations clearly explained) which is the dominant approach for other types of spin pumping problems in spintronics..
We agree with the referee and have added a third paragraph to explain this issue. In particular we state
"In this manuscript, we exploit the time- and space-tunability of Rashba SOI to ex- plore how the non-Abelian (spin) gauge fields induce spin transport. We stress from the outset that the possibility of using time-dependent Rashba SOI for generating (i) AC spin currents in bulk diffusive systems [8, 27] and (ii) DC spin currents (hence spin pump- ing) in mesoscopic quantum wires [9] and quantum dots [10] has already been proposed. Moreover Onsager reciprocity relations have also been extended to spin dependent trans- port [16, 28, 29]. However, it was understood later on that in the presence of an additional non-Abelian gauge structure such as the one induced by the space- and time-dependent Rashba interaction, Onsager relations become more restrictive [17]. Here, we further explore the consequences of such ``hidden'' Onsager relations recently established for a spin and charge transport in 2DEGs. Among other things, our approach is applicable to diffusive and ballistic systems and is capable of handling effects of quantum coherence. Moreover, our method allow us to obtain general analytical formulae that go beyond the frozen scattering matrix approximation. [30,31] –the conventional approach for spin pumping problems [32]."
- Equation 34 is written without any citation, which gives impression that the authors have derived it in the present manuscript. However, this seems to be just usage of Equations from Sec. 3.3.1 of Moskalets' book, cited as Ref. 23, so the authors should add appropriate citations here and in Appendices.
We thank the referee for pointing this to us. We added the citations at relevant places: In the section 3.2.1 before Eq.34 (Eq.35 in the latest version) and in the appendix D before Eq.70.
- The authors evaluate Eq. 34 using just one harmonic. However, very recent studies: ...point out that time-dependent Rashba systems invariably generate higher than one harmonic in pumped spin and/or charge currents. Thus, this is my MAIN suggestion which could CONNECT paper with this contempory direction -> try to compute higher harmonics in pumped current. Also, as discussed in B. above, once the device is left-right asymmetric [see also Fig. 4 in J. Phys.: Mater. 2, 025004 (2019) or Phys. Rev. B 72, 245339 (2005) for detailed discussion about symmetry breaking requirements], one should observe DC charge current without any second "2DEG converted" in Fig. 1. I am wondering if the authors have checked that by simply computing Eq. 34 for charge current.
We thank the referee for the interesting suggestion and the references. Indeed, the referee is right that there is a DC charge current in the system which we computed using Floquet scattering matrix. This DC charge current is small compared to the AC charge current. The reason is that the DC contribution is proportional to the square of the driving frequency as a result of having a single time dependent parameter in our system [1]. We have added a discussion about the DC current in section 4.3.
We also investigated higher harmonic generation of the spin and charge currents. We found that spin currents in x and y directions receive contributions from odd harmonics. The spin current in z direction and the charge current receive contributions from the even ones. We have added our results in a new section, Appendix G.
- The definition of adiabatic limit, "If the frequency is much smaller than the inverse of the time of fight, ..., the scattering process is considered to be in the adiabatic limit.", does not take into account the relationship between frequency and Fermi energy. That is, one can see immediately from Eq. 34 that integration is not needed, which is adiabatic, if frequency is much smaller than the Fermi energy. Some comment on the connection between these definitions would be useful.
We modified the sentence to address the referee’s comment by including the relation to the Fermi energy in the definition of the adiabatic limit. It now reads "We will consider a wide range of frequencies, numerically probing both the low-frequency (${\hbar\Omega}/(\hbar/\tau$) \ll 1 $) and high-frequency regimes (${\hbar\Omega}/(\hbar/\tau$) \gtrsim 1 $), where $\hbar/\tau$ is the typical internal energy scales of the scattering matrix. For few channel ballistic transport, $\tau$ is the time of flight of an electron between two leads, which is calculated using the Wigner-Smith time-delay matrix \cite{wigner, smith} as shown in Appendix \ref{sec.time_f}. If the frequency is much smaller than the inverse of the time of flight and the Fermi energy $\epsilon_{F}$, $\Omega \tau \ll 1$ and $\hbar\Omega \ll \epsilon_{F}$, the scattering process is in the adiabatic limit."
- In Fig. 9 it will look puzzling to experts in general theory of Floquet systems that the results are NOT very sensitive to the driving frequency (left versus right panels). Usually it is the combination of both the driving frequency and amplitude which sets the regime.
We thank the referee for this comment which helped us improve the clarity of the Figure 9 (Figure 10 in the latest version). To address this issue, we added an explanation on how the cut-off value of the Floquet channel number depends on the driving frequency as well as the amplitude of the time dependent potential. In particular, we state, in the appendix F : "The ratio between the amplitude of the oscillating potential and the driving frequency determines the cut-off value $n_max$. As discussed in Ref.33, if the amplitude of the oscillating potential is much smaller than ~\hbar\Omega, $n_max$ = 1 is sufficient for numerical calculations. On the other hand, if the amplitude of the oscillating potential is comparable to the driving frequency, one needs $n_max$ > 1 for numerical calculations." and "Generally, we observe that smaller cut-off values $n_{max}$ suffice for smaller choices of $k_{so}$, independent of frequency, as expected. At moderate values of $k_{so}$, while smaller $n_{max}$ suffice for smaller frequencies, higher $n_{max}$ is needed at higher frequencies."
- The authors repeatedly use term "spin-motive force", including in abstract and conclusion: However, this term is widely accepted to denote CHARGE pumping by dynamical noncollinear and noncoplanar magnetic textures (such as domain walls)...So, to avoid confusion, they should change this terminology.
We changed the term “spin-motive force” to “spin electric force” in two occasions in the draft.
- Current manuscript has many typos: under Eq. 52 "prediction demostrating"; "Figure 7: Dyanmical spin transistor", ... Please run spell checker.
We thank the referee for pointing this out.
Attachment:
Dynamical_Spin_Orbit_Based_Spin_Transistor_.withcorrections.pdf
Bertrand Berche on 2022-04-03 [id 2351]
This is a comment on section 2 of the paper, which appeals to me because of related works that I have been able to carry out in collaboration, in particular with E Medina.. The authors consider here a 2DEG with a time or space-dependent Rashba SOI i.e. with an amplitude $\alpha_R({\bf x},t)$ (eqn (6)). The theory has the structure of an SU(2) gauge theory.
When the amplitude $\alpha_R$ only depends on $t$, a gauge transformation maps the system on a spin-dependent electric field (an electric field which accelerates opposite spin species in opposite directions). When $\alpha_R$ is space-dependent only, the SU(2) gauge mapping leads to a magnetic field acting with opposite Lorentz forces on the two spin species. This sounds very good, but I have a question regarding the gauge transformation and the nature of the gauge transformed problems.
A first observation (see arXiv:0902.4635 and arXiv:1202.4085) is that the non relativistic 2DEG has no gauge freedom owing to the quadratic term in the SOI amplitude (this is similar to the diamagnetic gauge fixing term in superconductivity). This problem can be overcome if one neglects the second order terms in the Hamiltonian, or for example in the relativistic case, relevant to graphene Physics. But there is still another difficulty: The Rashba amplitude is given in terms of the electric field (intrinsic or external) acting on the 2DEG. This is hidden in the spin-orbit field ${\bf b}({\bf p})$. The non-Abelian gauge vectors are given in terms of $\alpha_R$, hence in terms of the electric field. As a consequence, one cannot proceed to a SU(2) gauge transformation without changing the physical situation. This has been exploited e.g. in arXiv:1506.06571, where it was shown that the SU(2) gauge transformations map different physical situations onto each other.
I am wondering how the results obtained in the present paper remain robust with respect to this difficulty.

---

## Round 1 · Referee Report · Anonymous · 2022-4-7

Strengths
The paper deals with important topics in Spintronics, that is all electrical generation of spin currents and the design of a robust and experimentally realisable spin transistor.
1) The paper is well written and sufficiently self-contained that an expert in the field can follow all derivations without the need to peruse other material.
2) The physics at play is clearly explained.
3) The results from the analytical theoretical treatments are validated by a full numerical approach. The results are plausible and in my opinion correct.
Weaknesses
1. The main weakness of this manuscript is that it is not easy to distinguish which ideas or results are novel.
2. The paper proposes a design for a spin transistor. However, no result is shown for the transistor operation, that is the on/off switching of the current.
Report
The authors present a detailed study of a mesoscopic spin transistor realised in a two-dimensional electron gas exploiting the tunability of the Rashba spin-orbit interaction (SOI). The device functionality relies on the generation of a spin current by a time-dependent Rashba SOI and by its transformation into a charge current in a region with spatially inhomogeneous SOI in a three-terminal geometry. The authors exploit the fact that the SOI can be written as a non-Abelian gauge field to derive analytical expressions for the charge and spin currents. These expressions are then compared with a full numerical calculation.
Below, I outline my main comments on the paper:
1. In my opinion, in this paper it is difficult for the reader to distinguish which results are new and what is already known in the literature. For example, the idea of exploiting a time-dependent SOI to generate a spin current was presented in PRB 68, 154324 (2003) and PRB 68, 233307 (2003). In the first of these two papers the spin-current generation is also described in terms of a spin-dependent electric field. The fact that there are previous works in the literature, is not at all clear in Sections 4 and 2.2 . Similarly, the idea of exploiting an inhomogeneous SOI to transform spin currents into charge currents has already been presented in Ref. [16] and it would be useful for the reader to have this clearly stated at the beginning of Section 5.
2. In Section 5, the authors describe a system that combines the generation of a time-dependent spin current and its detection.
2.(a) It is not clear to me why the authors consider a Aharonov-Bohm ring geometry in the detection part of the setup. This would be understandable if they were using a magnetic field to break time reversal symmetry and achieve spin detection. Without magnetic field the ring seems only an unnecessary complication.
2(b) Since the author emphasise that the device is a "dynamical spin-orbit based spin transistor", they need to show that the system indeed works as a transistor. In particular, they state "controlling the symmetry properties of the system, we can obtain on/off states of this dynamical spin transistor." The authors need to provide a figure where this on/off switching is shown as a function of some external parameter (perhaps, a tunnel coupling to lead 2).
3) In the captions of Figs 3,4, 5 and 7 . The author should state that the current is plotted as a function of the Fermi Energy.
Requested changes
1) Add citations and if relevant modify the text to make it easier to distinguish new results from results already known, as detailed in point 1 of the report.
2) Introduce a figure in Section 5 , showing the on/off switching of the device.
3) Improve the figure captions (see, point 3 in the report).
Author: Fahriye Nur Gursoy on 2022-10-06 [id 2892]
(in reply to Report 2 on 2022-04-07)
We thank the referee for their report. It seems the referee's main criticism is that
In my opinion, in this paper it is difficult for the reader to distinguish which results are new and what is already known in the literature.
In order to make this point clear, we have added a new paragraph to the Introduction. In particular, we stress that our numerical work generalizes earlier work to fully quantum coherent systems of arbitrary shape and applies in both clean and disordered/dirty limits, even if our focus is on ballistic devices. Moreover, our analytical results apply beyond the range of validity of frozen scattering matrix approximation --the conventional method to analyze these time dependent systems.
For example, the idea of exploiting a time-dependent SOI to generate a spin current was presented in PRB 68, 154324 (2003) and PRB 68, 233307 (2003). In the first of these two papers the spin-current generation is also described in terms of a spin-dependent electric field. The fact that there are previous works in the literature, is not at all clear in Sections 4 and 2.2 . Similarly, the idea of exploiting an inhomogeneous SOI to transform spin currents into charge currents has already been presented in Ref. [16] and it would be useful for the reader to have this clearly stated at the beginning of Section 5.
We thank the referee for pointing this reference to us (we presume it is PRB 68, 155324 (2003)). We have added citations to it in Secs.2, 4 and 5.
2.(a) It is not clear to me why the authors consider a Aharonov-Bohm ring geometry in the detection part of the setup. This would be understandable if they were using a magnetic field to break time reversal symmetry and achieve spin detection. Without magnetic field the ring seems only an unnecessary complication.
The interference between paths in a ring geometry is sensitive to both SU(2) and the usual U(1) phases. Moreover, the opening of a third lead in the ring allows us to bypass the restrictions imposed by Onsager relations for a two lead system. All this leads to an enhanced on/off ratio (current Fig.8), where the off state is induced by pinching off the third lead. If the ring is exchanged with another shape our results stay the same albeit with a smaller on/off ratio.
2(b) Since the author emphasise that the device is a "dynamical spin-orbit based spin transistor", they need to show that the system indeed works as a transistor. In particular, they state "controlling the symmetry properties of the system, we can obtain on/off states of this dynamical spin transistor." The authors need to provide a figure where this on/off switching is shown as a function of some external parameter (perhaps, a tunnel coupling to lead 2).
We thank the referee for reminding this to us. We have added Fig.8, explicitly showing the transistor action.
3) In the captions of Figs 3,4, 5 and 7 . The author should state that the current is plotted as a function of the Fermi Energy.
The captions of the Fig 3, 4, 5 and 7 are updated according to referee’s comment.
Attachment:
Dynamical_Spin_Orbit_Based_Spin_Transistor_.withcorrections_Xh71B4D.pdf

---

## Round 1 · Referee Report · Anonymous · 2022-4-13

Strengths
1-Comprehensive and unified description of non-Abelian gauge-field approach to calculating spin-dependent transport.
2-Striking comparison of analytical predictions with numerical Floquet simulations.
3-The authors discuss an interesting device-application proposal.
Weaknesses
1-Acknowledgement of previous work on spin currents arising from time-varying and spatially inhomogeneous spin-orbit coupling is incomplete.
2-Some crucial aspects of the new formalism are discussed only cursorily.
Report
The authors present an interesting extension of the formalism that views linear-in-momentum spin-orbit coupling as a spin-dependent non-Abelian gauge field. They derive analytical results for spin currents arising from time-varying and/or spatially inhomogeneous spin-orbit-coupling strengths. Numerical results based on the powerful Floquet scattering theory support the applicability of the new theory well beyond the adiabatic limit that has been the focus of previous spin-dependent mesoscopic-transport theories.
The present work constitutes a follow-up on earlier results by some of the authors reported in Ref. [16]. As such, one would expect some more detail in the present manuscript regarding certain basic derivations that had to be omitted in the short letter publication [16]. In particular, Sec. 3.1.2 feels short and at times cryptic, leaving especially unclear the type and level of approximations involved in the derivations. I would suggest the extend the discussion to include more detailes and rigorous statements about validity of the approximations employed.
The new approach utilized by the authors is more generally applicable than previously used scattering-theory formalisms. While it is useful to demonstrate clearly the deviation between the two theories by direct comparison for particular parameters as is done in Fig. 5, it would be even more useful and interesting to the reader to understand better the appropriate limit in which the adiabatic result is contained within the more general approach. The discussion of Sec. 3.2.2 is again quite short in this regard, merely stating the almost obvious fact that corrections are of order Omega. But what is the scale to compare Omega with? The authors may want to clarify this.
Finally, I think it would be justified to acknowledge a few key earlier contributions to the understanding of spin-orbit coupling acting like a gauge field, and its time dependence generating spin-dependent electric fields. In addition to current Ref. [9], the paper by Governale, Taddei and Fazio [PRB 68, 155324 (2003)] was seminal. Also, Avishai, Cohen and Nagaosa [PRL 104, 196601 (2010)] made an important contribution. Potentially there are more relevant citations, especially from the spin-pumping literature, that would be appropriately included in the paper's reference list. In addition to merely giving the references, it would also be useful to state more explicitly (maybe in the conclusions?) which qualitatively new effects and additional transport regimes the current manuscript's formalism allows to explore compared to the earlier types of scattering formalism. Furthermore, clarifying briefly also in the part discussing the "gauging away" of constant spin-orbit coupling [paragraph below Eq. (5)] which types of transport effects still occur at order k_so^2 (weak antilocalization!?) would be helpful to better embed the current approach to prior ones.
Requested changes
See my report.
Author: Fahriye Nur Gursoy on 2022-10-06 [id 2893]
(in reply to Report 3 on 2022-04-13)
We thank the referee for their report. We address their comments point by point below. It seems the main criticism of the referee is
Acknowledgement of previous work on spin currents arising from time-varying and spatially inhomogeneous spin-orbit coupling is incomplete.
We have added a new paragraph in the Introduction to discuss this point and how these works relate to the present work.
The present work constitutes a follow-up on earlier results by some of the authors reported in Ref. [16]. As such, one would expect some more detail in the present manuscript regarding certain basic derivations that had to be omitted in the short letter publication [16]. In particular, Sec. 3.1.2 feels short and at times cryptic, leaving especially unclear the type and level of approximations involved in the derivations. I would suggest the extend the discussion to include more detailes and rigorous statements about validity of the approximations employed.
All our analytical results are valid to linear order in the spin orbit coupling constant. We have rewritten sec.3.1.2 in order to better explain the steps in our approximation scheme.
The new approach utilized by the authors is more generally applicable than previously used scattering-theory formalisms. While it is useful to demonstrate clearly the deviation between the two theories by direct comparison for particular parameters as is done in Fig. 5, it would be even more useful and interesting to the reader to understand better the appropriate limit in which the adiabatic result is contained within the more general approach. The discussion of Sec. 3.2.2 is again quite short in this regard, merely stating the almost obvious fact that corrections are of order Omega. But what is the scale to compare Omega with? The authors may want to clarify this.
We agree with the referee and thank them for pointing this to us. We now explicitly state the conditions for the validity of the adiabatic approximation just before section 3.2.2.:
"We will consider a wide range of frequencies, numerically probing both the low-frequency (${\hbar\Omega}/(\hbar/\tau$) \ll 1 $) and high-frequency regimes (${\hbar\Omega}/(\hbar/\tau$) \gtrsim 1 $), where $\hbar/\tau$ is the typical internal energy scales of the scattering matrix. For few channel ballistic transport, $\tau$ is the time of flight of an electron between two leads, which is calculated using the Wigner-Smith time-delay matrix \cite{wigner, smith} as shown in Appendix E. If the frequency is much smaller than the inverse of the time of flight and the Fermi energy $\epsilon_{F}$, $\Omega \tau \ll 1$ and $\hbar\Omega \ll \epsilon_{F}$, the scattering process is in the adiabatic limit."
Finally, I think it would be justified to acknowledge a few key earlier contributions to the understanding of spin-orbit coupling acting like a gauge field, and its time dependence generating spin-dependent electric fields. In addition to current Ref. [9], the paper by Governale, Taddei and Fazio [PRB 68, 155324 (2003)] was seminal. Also, Avishai, Cohen and Nagaosa [PRL 104, 196601 (2010)] made an important contribution. Potentially there are more relevant citations, especially from the spin-pumping literature, that would be appropriately included in the paper's reference list.
We thank the referee for pointing these references to us. We have included them, together with a comment on the more general spin-pumping literature and higher harmonics. However, we would like to stress that our focus here is the AC spin/charge currents rather than the DC currents that would be produced via spin pumping. For the systems we consider in the present manuscript, the AC charge/spin currents are an order of magnitude larger than the pumping currents.
In addition to merely giving the references, it would also be useful to state more explicitly (maybe in the conclusions?) which qualitatively new effects and additional transport regimes the current manuscript's formalism allows to explore compared to the earlier types of scattering formalism. Furthermore, clarifying briefly also in the part discussing the "gauging away" of constant spin-orbit coupling [paragraph below Eq. (5)] which types of transport effects still occur at order k_so^2 (weak antilocalization!?) would be helpful to better embed the current approach to prir ones.
We would like to point out that all quantum coherent effects are included in our numerical simulations. For instance AB/AC like quantum interference oscillations form the basis of high on/off ration for our spin transistor proposal in Fig.6. The SOI is taken into account up to order (k_SO L) in our analytical formulae, hence effects such as weak antilocalization do not yet form up fully in the charge transport. Extending our method including analytics to higher order in the spin-orbit interaction would clearly be interesting, but beyond the scope of the present manuscript.
Attachment:
Dynamical_Spin_Orbit_Based_Spin_Transistor_.withcorrections_anBJCBz.pdf

---

## Round 4 · Referee Report · Anonymous (Referee 3) · 2022-10-15

Report

In the resubmitted manuscript, the authors have satisfactorily addressed criticisms and recommendations voiced in the referee reports on the previous version. I am therefore happy to recommend acceptance in SciPost Physics.

---

## Round 4 · Referee Report · Anonymous (Referee 1) · 2022-11-28

Report

The authors have clarified different issues raised by the referees, improved citations and some of the results. So, the manuscript can be published in it present form.

---

## Editorial Decision

published